# Structural insights into the mechanism of the DEAH-box RNA helicase Prp43

**Marcel J Tauchert[1], Jean-Baptiste Fourmann[2], Reinhard Lührmann[2], Ralf Ficner[1]\***

[1]Department of Molecular Structural Biology, Institute for Microbiology and Genetics, GZMB, Georg-August-University Göttingen, Göttingen, Germany; [2]Department of Cellular Biochemistry, Max Planck Institute for Biophysical Chemistry, Göttingen, Germany

**Abstract** The DEAH-box helicase Prp43 is a key player in pre-mRNA splicing as well as the maturation of rRNAs. The exact *modus operandi* of Prp43 and of all other spliceosomal DEAH-box RNA helicases is still elusive. Here, we report crystal structures of Prp43 complexes in different functional states and the analysis of structure-based mutants providing insights into the unwinding and loading mechanism of RNAs. The Prp43•ATP-analog•RNA complex shows the localization of the RNA inside a tunnel formed by the two RecA-like and C-terminal domains. In the ATP-bound state this tunnel can be transformed into a groove prone for RNA binding by large rearrangements of the C-terminal domains. Several conformational changes between the ATP- and ADP-bound states explain the coupling of ATP hydrolysis to RNA translocation, mainly mediated by a $\beta$-turn of the RecA1 domain containing the newly identified RF motif. This mechanism is clearly different to those of other RNA helicases.

## Introduction

Helicases are omnipresent enzymes, spread through the whole phylogenetic tree of life as well as viruses, since their unwinding capabilities are of striking importance for numerous cellular pathways of RNA and DNA metabolism. Based on sequence alignments, helicases were subdivided into six superfamilies (SF) which come in two different flavors (*Singleton et al., 2007*; *Fairman-Williams et al., 2010*; *Jankowsky and Fairman, 2007*; *Pyle, 2011*). Most members of the SF1 or SF2 are monomeric, in contrast to the members of SF3 to SF6 which assemble as hexamers (*Singleton et al., 2007*). The latter are composed of either RecA- or AAA$^+$-like domains, thus being distinct from the SF1 and SF2 helicases which are exclusively composed of two RecA-like domains in juxtaposition with each other.

The SF2 is by far the largest superfamily, comprising a vast number of RNA helicases with implications in numerous RNP metabolism pathways, such as translation initiation and termination, pre-mRNA editing and splicing, mRNA export as well as ribosome biogenesis (*Ozgur et al., 2015*). The centerpiece of these helicases is formed by the two RecA-like domains, harboring conserved motifs which are crucial for these NTPase-dependent RNA helicases to accomplish their function. The motifs I, II, V and VI are required for nucleoside triphosphate binding and hydrolysis, motifs Ia, Ib and IV are involved in RNA binding and the motif III participates in coupling the NTPase and the unwinding activity (*Caruthers and McKay, 2002*; *Cordin and Beggs, 2013*; *Cordin et al., 2012*). Most SF2 helicases exhibit extensions at the N- and C-termini of the helicase core, which is also true for the DEAH-box proteins, a subfamily of the DEAH/RHA family. The C-terminal extension is highly conserved in DEAH-box helicases and is composed of three domains: a winged helix (WH), a ratchet and an oligosaccharide binding (OB) fold (*Walbott et al., 2010*; *He et al., 2010*). The ratchet domain was originally named according to the homologous domain of the DNA helicase Hel308

*For correspondence: rficner@ uni-goettingen.de

**Competing interests:** The authors declare that no competing interests exist.

(*Büttner et al., 2007*), but recent studies on MLE suggested that this domain has no ratcheting function and should be renamed (*Prabu et al., 2015*). Therefore it is hereinafter referred to as ratchet-like domain. In DEAH-box proteins the degrees of freedom of the helicase core are reduced by the interactions between the C-terminal domains and the two RecA-like domains in comparison to those DEAD-box proteins lacking such C-terminal domains. The conservation of the N-terminal extension is distinctly lower and they dramatically differ in length between the members of this subfamily. Reported functions for this N-terminal extension are, among others, the involvement in the recruitment as shown for Prp16 (pre-mRNA processing factor 16) as well as for Prp22 or for the subcellular targeting to the nucleus as elucidated for Prp43 (*Wang and Guthrie, 1998*; *Schneider and Schwer, 2001*; *Fouraux et al., 2002*).

The latter mentioned helicase, Prp43, is an outstanding member of the DEAH-box subfamily since it has implications in different substantive cellular processes. The first reported function of Prp43 is its involvement in pre-mRNA splicing. In the course of this process, Prp43 acts at the latest stage of the splicing cycle and it is required to dismantle the intron-lariat spliceosome into the excised lariat and the U2•U5•U6 snRNPs (*Arenas and Abelson, 1997*; *Fourmann et al., 2013*, *2016a*). Recently, the target substrate of Prp43 during this process was revealed which is the RNA network between the U2 snRNP and the branch site of the intron (*Fourmann et al., 2016b*). Indeed, in the spliceosome, Prp43 was crosslinked exclusively to the pre-mRNA and not to any snRNAs. Furthermore, Prp43 is also a key player for the disassembly of stalled spliceosomes (*Pandit et al., 2006*; *Semlow and Staley, 2012*; *Koodathingal et al., 2010*). During these spliceosomal processes, Prp43 binds RNA in a sequence-independent manner. Prp43 action is also essential for ribosome biogenesis. Here, it is involved in the maturation of 18S and 25S rRNAs with several reported binding sites on intermediates which vary in sequence and thus make a sequenced-based binding mechanism unlikely for Prp43 (*Lebaron et al., 2005*; *Bohnsack et al., 2009*). Lately, the involvement of Prp43 in a third distinct cellular function was illuminated and it was unraveled that Prp43 is also involved in the promotion of apoptosis (*Heininger et al., 2016*).

The distribution of Prp43 between these three fundamentally different processes is regulated by a fine-tuned interplay of the helicase itself and cofactors. These cofactors belong to the family of G-patch proteins, which share an intrinsically disordered glycine-rich region, the G-patch motif (*Aravind et al., 1999*). Apart from being involved in recruitment, these G-patch proteins also increase the NTPase and stimulate the helicase activity of Prp43. To date, in *S. cerevisiae* five G-patch proteins were identified out of which four interact with Prp43. In the context of splicing, Prp43 function is modulated by Ntr1 (Nineteen complex-related proteins 1) (*Tanaka et al., 2007*; *Tsai et al., 2005*, *2007*; *Boon et al., 2006*). During rRNA maturation, Prp43 is associated with Pfa1 (Prp forty-three associated 1) or Gno1 (G-patch nucleolar protein 1) (*Lebaron et al., 2005*, *2009*; *Bohnsack et al., 2009*). Moreover, its apoptotic function is implemented by the interaction with Cmg1 (Cytoplasmic and mitochondrial G-patch protein 1) (*Heininger et al., 2016*). The remaining G-patch protein from yeast is Spp2 (Suppressor of PRP protein 2) which interacts with another spliceosomal DEAH-box helicase, Prp2 (*Warkocki et al., 2015*).

Despite intensive studies, the exact *modus operandi* of the DEAH-box helicases and thus of Prp43 is still elusive to date. This is primarily attributable to a lack of crystal structures of members of the DEAH-box subfamily in an active state and with bound interaction partners. Recently, the crystal structure of the related DExH-box helicase MLE (Maleless), which belongs to another DEAH/RHA subfamily, provided a first insight into how this helicase might function. The complex of MLE with a $U_{10}$-RNA and ADP•AlF$_4^-$ led to a proposed mechanism of how RNA is translocated through its interior tunnel driven by conformational changes of a Hook-Loop in the RecA2 domain (*Prabu et al., 2015*).

So far, three crystal structures of Prp43 in the post-catalytic ADP-bound state have been available from *S. cerevisiae* or *C. thermophilum* (*Walbott et al., 2010*; *He et al., 2010*; *Tauchert et al., 2016*). The latter mentioned organism might be a valuable alternative for structural studies on DEAH-box proteins, since several crystal structures of spliceosomal RNA helicases from this thermophilic fungus were solved only recently (*Tauchert et al., 2016*; *Tauchert and Ficner, 2016*; *Absmeier et al., 2015*). Owing to this, intensive crystallization trails with Prp43 from *C. thermophilum* were performed leading to three new crystal structures. All three structures show Prp43 in an active state, since it was crystallized with a non-hydrolysable ATP-analog (ADP•BeF$_3^-$) as well as one of these structures show Prp43 with an additional $U_7$-RNA. The Prp43•$U_7$-RNA•ADP•BeF$_3^-$ complex

unveils the structural basis of why Prp43 can bind RNA in a sequence-independent manner. The Prp43•ADP•BeF$_3^-$ structures illustrate how Prp43 binds and releases complex RNA substrates due to massive rearrangements of the C-terminal domains which affect an opening of the interior RNA-binding tunnel which is formed by the two RecA-like and the C-terminal domains. The hydrolysis of an ATP molecule induces conformational rearrangements of the RecA-like domains which, in turn, allow the translocation of the RNA through the interior tunnel *via* the so far unrecognized RF motif present in a $\beta$-turn of all DEAH-box helicases. According to our results, the translocation mechanism of MLE is very unlikely to be compatible for Prp43. All proposed functional mechanisms of Prp43 are strengthened by functional analyses of structure-based mutants in this study.

## Results

### Crystal structure of Prp43 with bound RNA in an active state

In order to overcome the lack of crystal structures of DEAH-box helicases in an active state, we crystallized Prp43 in the presence of the non-hydrolysable ATP-analog ADP•BeF$_3^-$ as well as a U$_{16}$-RNA. For these crystallization approaches, an N-terminally truncated version of Prp43 from *C. thermophilum* was used (ctPrp43$\Delta$N), which lacks the first 60 amino acids (*Figure 1a*). Previously, we already demonstrated that this truncated ctPrp43 variant is fully capable of replacing its yeast ortholog in spliceosome disassembly assays (*Tauchert et al., 2016*). Crystals of ctPrp43$\Delta$N•ADP•BeF$_3^-$ in the presence of the U$_{16}$-RNA were obtained in the space group P6$_1$22 and diffracted up to 2.62 Å. In addition to the ADP•BeF$_3^-$ and a central magnesium ion at the active site, seven out of the 16 nucleotides of the RNA as well as one additional phosphate group were traceable in the electron density (for an omit map of the RNA see *Figure 1—figure supplement 1a*). Hence, this complex is referred to here as ctPrp43$\Delta$N•U$_7$•ADP•BeF$_3^-$. The refinement of this complex converged at R$_{work}$ and R$_{free}$ values of 19.70% and 22.97% (*Table 1*).

By this ctPrp43$\Delta$N•U$_7$•ADP•BeF$_3^-$ complex structure (*Figure 1b–1e*), insights into the sequence-unspecific RNA-binding mechanism of a genuine DEAH-box helicase were obtained. ctPrp43 consists of six different domains: an N-terminal extension (residues 1–96), the RecA1 (97–273) and RecA2 (274–458) domains, and the C-terminal domains which were assigned as degenerate winged helix domain (459–526), ratchet-like domain (527–640) and oligosaccharide binding-fold domain (641–764) (*Walbott et al., 2010*; *He et al., 2010*).

The RecA-like domains and the C-terminal domains appear to be stably associated by multiple interactions and form a tunnel inside the Prp43 molecule (*Figure 1d*) which is the binding site of the U$_7$-RNA. The 5' end of the RNA is located at the RecA2 domain and the 3' RNA end is situated at the RecA1 domain. For the first three nucleotides at the 5' end (U1–U3) two alternative conformations are observed (*Figure 1—figure supplement 2*). The structure reveals the basis for the sequence-independent RNA binding of Prp43 which is in line with previous biochemical data (*Fourmann et al., 2016b*; *Bohnsack et al., 2009*; *Tanaka and Schwer, 2006*).

### Prp43 binds RNA in a sequence-independent fashion

Interactions between Prp43 and the U$_7$-RNA are sparsely present. Depending on which of the two alternative RNA conformations is analyzed, only 11 or 13 hydrogen bonds and ionic interactions between the RNA and Prp43 can be detected (*Figure 1c and e*). Most of these interactions are formed between the sugar-phosphate backbone of the RNA and the RecA1, RecA2 or the ratchet-like domain. On the contrary, the uracil moieties do not interact with the helicase with the exception of a hydrogen bond of U3 with the Ser 555 main-chain carboxyl group from the ratchet-like domain and a base stack between the U1 and Arg 562 which only occurs in one of the two alternative conformations. These Prp43-RNA interactions are, however, by far not enough for sequence-specific binding of a substrate RNA to Prp43. In addition to the unspecific RNA-binding mode of Prp43, RNA binding appears to be transient, as only four nucleotides are more tightly associated with Prp43. In the crystal structure presented, this corresponds to the nucleotides U4 to U7. At the 5' end, the first three nucleotides are weakly bound by Prp43 since only 2 (conf. B) and 3 (conf. A) interactions are detectable, respectively. The weak interaction is additionally highlighted by elevated *B*-factors for this RNA region compared to the four other nucleotides (see *Figure 1—figure supplement 3*). After the first two nucleotides, a kink is introduced in the RNA backbone and the number

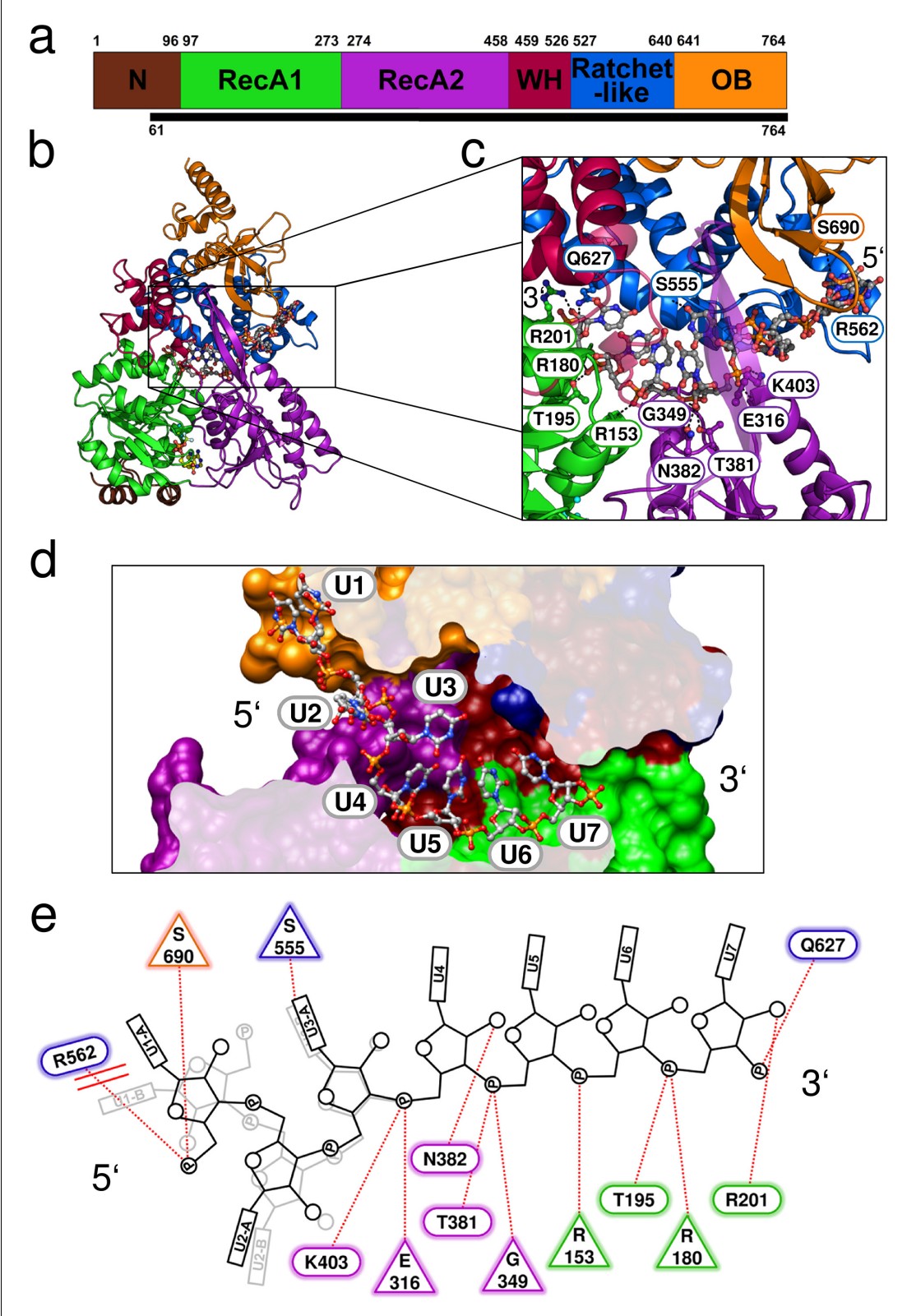

**Figure 1.** Crystal structure of Prp43 in complex with $U_7$-RNA and the ATP-analog ADP•BeF$_3^-$. (a) Domain overview of ctPrp43. The bottom bar indicates the N-terminally truncated construct ($\Delta$N, 61–764) used for crystallization. (b) Overall structure of ctPrp43$\Delta$N•$U_7$•ADP•BeF$_3^-$. Domains are colored according to a and the bound $U_7$-RNA is shown in gray with two alternative conformations for nucleotides U1-U3. The ADP•BeF$_3^-$ is bound in the cleft between the RecA domains. (c) Close-up of the bound $U_7$-RNA. Residues involved in interactions are labeled according to the wild-type ctPrp43

*Figure 1 continued on next page*

Figure 1 continued

sequence. The 5' and 3' end of the RNA is indicated. (d) Cross-section of the Prp43 RNA-binding tunnel. Prp43 is shown in surface representation and the RNA in ball-and-sticks mode. (e) Schematic figure of the Prp43-RNA interactions. Residues which interact with the RNA *via* their main chain are shown as triangles and residues which exhibit side chain interactions are presented as ellipses. The coloring of the residues corresponds to a. The alternative conformation of the first three nucleotides is shown light gray. Stacking interactions are highlighted by double lines and polar interactions by dotted lines.

The following figure supplements are available for figure 1:

**Figure supplement 1.** Omit maps of a fraction of the U$_7$-RNA from the ctPrp43ΔN•U$_7$•ADP•BeF$_3^-$ complex structure and of the active site of ctPrp43ΔN•ADP•BeF$_3^-$(HR) (*Figure 2a*).

**Figure supplement 2.** The two alternative conformations of the U$_7$-RNA in the ctPrp43ΔN•U$_7$•ADP•BeF$_3^-$ complex structure.

**Figure supplement 3.** Overview of the *B*-factors of the U$_7$-RNA in the ctPrp43ΔN•U$_7$•ADP•BeF$_3^-$ complex structure.

**Figure supplement 4.** Schematic representation of the NS3 HCV- and MLE-RNA interaction networks.

of Prp43-RNA interactions gets more numerous and more intense. At the 3'end, the RNA does not interact with Prp43 as soon as the end of the RNA-binding tunnel is reached. The phosphate group of the eighth nucleotide is exactly located at the end of this tunnel. Compared to the nucleotides U4 to U7, this phosphate exhibits elevated *B*-factors. The U8 nucleoside was not defined in the electron density map and thus might indicate the presence of multiple RNA conformations at this site out of which none is clearly favorable due the lack of interactions with Prp43.

Having determined the crystal structure of ctPrp43ΔN•U$_7$•ADP•BeF$_3^-$ another question concerning Prp43's *modus operandi* did arise which is directly linked to its cellular role. In the context of pre-mRNA splicing or rRNA biogenesis, Prp43 has to bind to or to be released from short singled-stranded RNA stretches within larger folded RNAs or RNPs. In the light of the fact that the RNA-binding site of Prp43 is located inside a tunnel, RNA loading appears to be non-trivial and thus requires further illumination.

## Crystal structure of Prp43 in the ATP-bound state

In pursuance of analyzing how Prp43 binds to ssRNA regions within folded RNAs, we targeted at determining the pre-catalytic state of Prp43 before RNA binding. For this purpose, we subjected ctPrp43ΔN to crystallization trials in the presence of the ATP analog ADP•BeF$_3^-$. The ctPrp43ΔN•ADP•BeF$_3^-$ complex crystallized in two very distinct crystallization conditions and in two different space groups. The structure of the complex, which crystallized in P2$_1$2$_1$2$_1$, was refined at a resolution of 1.78 Å to R$_{work}$ and R$_{free}$ values of 17.67% and 19.88% (*Figure 2a*) and is referred to here as ctPrp43ΔN•ADP•BeF$_3^-$(HR) (high resolution). The second crystal form belongs to the hexagonal space group P6$_5$ which exhibited a resolution limit of 3.24 Å (*Figure 2b*) and thus is referred to as ctPrp43ΔN•ADP•BeF$_3^-$(LR) (low resolution). This structure was refined to R$_{work}$ / R$_{free}$ values of 18.22% and 21.78%, respectively.

In comparison to the ctPrp43ΔN•U$_7$•ADP•BeF$_3^-$ complex structure, the conformation of the two RecA-like domains in the ctPrp43ΔN•ADP•BeF$_3^-$ complex is almost identical and they superpose very well with r.m.s.d. values of 0.34 Å (312 C$_\alpha$) in case of ctPrp43ΔN•ADP•BeF$_3^-$(HR) and with 0.39 Å (318 C$_\alpha$) for the ctPrp43ΔN•ADP•BeF$_3^-$(LR) structure (*Figure 2c*). However, a large rearrangement of the C-terminal domains is seen in both ctPrp43ΔN•ADP•BeF$_3^-$ complex structures (*Figure 3*), which leads to the disruption of the interaction between two β-hairpin loops from the RecA2 domain and the OB-fold. As a consequence, the RNA-binding tunnel of ctPrp43ΔN•U$_7$•ADP•BeF$_3^-$ is opened and transformed into a shallow groove. In the closed state, the C-terminal domains of the ctPrp43ΔN•U$_7$•ADP•BeF$_3^-$ complex and of ctPrp43ΔN•ADP superpose well (r.m.s.d. of 0.55 Å for 267 C$_\alpha$).

**Table 1.** Data collection and refinement statistics.

| | ctPrp43ΔN•U$_7$•ADP•BeF$_3^-$ | ctPrp43ΔN•ADP•BeF$_3^-$(HR) | ctPrp43ΔN•ADP•BeF$_3^-$(LR) |
|---|---|---|---|
| PDBid | 5lta | 5ltj | 5ltk |
| **Data collection** | | | |
| Space group | P6$_1$22 | P2$_1$2$_1$2$_1$ | P6$_5$ |
| Cell dimensions | | | |
| a, b, c (Å) | 106.39, 106.39, 356.70 | 88.83, 105.64, 119.05 | 184.34, 184.34, 82.32 |
| α, β, γ (°) | 90.0, 90.0, 120.0 | 90.0, 90.0, 90.0 | 90.0, 90.0, 120.0 |
| Resolution (Å) | 48.56 – 2.62 (2.70 – 2.62) | 79.02 – 1.78 (1.89 – 1.78) | 92.17 – 3.24 (3.39 – 3.24) |
| R$_{meas}$ (%) | 5.9 (90.3) | 7.2 (123.5) | 9.8 (76.2) |
| I/σ(I) | 22.16 (1.85) | 14.88 (1.64) | 14.65 (2.26) |
| CC$_{1/2}$ (%) | 99.9 (65.2) | 99.9 (61.9) | 99.8 (63.8) |
| Completeness (%) | 98.5 (86.8) | 99.6 (98.8) | 98.8 (94.9) |
| Redundancy | 5.14 (5.34) | 4.71 (4.66) | 4.00 (3.59) |
| **Refinement** | | | |
| Resolution (Å) | 48.56 – 2.62 | 67.99 – 1.78 | 92.17 – 3.24 |
| No. reflections | 36887 | 107276 | 25304 |
| R$_{work}$ / R$_{free}$ | 19.70/22.97 | 17.67/19.88 | 18.22/21.78 |
| No. atoms | | | |
| Protein | 5605 | 5730 | 5622 |
| RNA | 204 | / | / |
| Ligand / Ion | 37 | 105 | 42 |
| Water | 52 | 688 | 4 |
| *B*-factors(Å$^2$) | | | |
| Protein | 72.83 | 34.41 | 88.87 |
| RNA | 88.65 | / | / |
| Ligand / Ion | 55.04 | 47.83 | 83.89 |
| Water | 60.51 | 44.29 | 62.19 |
| R.m.s. deviations | | | |
| Bond length (Å) | 0.0028 | 0.0041 | 0.0026 |
| Bond angles (°) | 0.78 | 0.87 | 0.67 |
| Ramachandran Plot | | | |
| Favored | 95.98 | 97.34 | 96.28 |
| Outlier | 0.0 | 0.0 | 0.0 |

Values in parentheses are for the highest resolution shell.

## RNA loading mechanism of Prp43

The opening of the RNA-binding tunnel appears to be the key mechanism allowing Prp43, and very likely numerous other DEAH-box helicases, to bind to complex folded RNA substrates. Thus, these rearrangements of the C-terminal domains are expected to be crucial for their unwinding function. The opening is feasible due to a large movement of the ratchet-like and the OB-fold domains. Rearrangements of the WH domain are less pronounced owing to the fact that the WH rather functions as a hinge region between the RecA2 and the ratchet-like domain. Comparison between the ctPrp43ΔN•U$_7$•ADP•BeF$_3^-$ and the ctPrp43ΔN•ADP•BeF$_3^-$(HR) complex structures after alignment of the RecA-like domains revealed that the center of mass of the ratchet-like domain is shifted by 16.2 Å, and by 14.1 Å for the OB-fold domain. In the ctPrp43ΔN•ADP•BeF$_3^-$(LR) complex structure, these

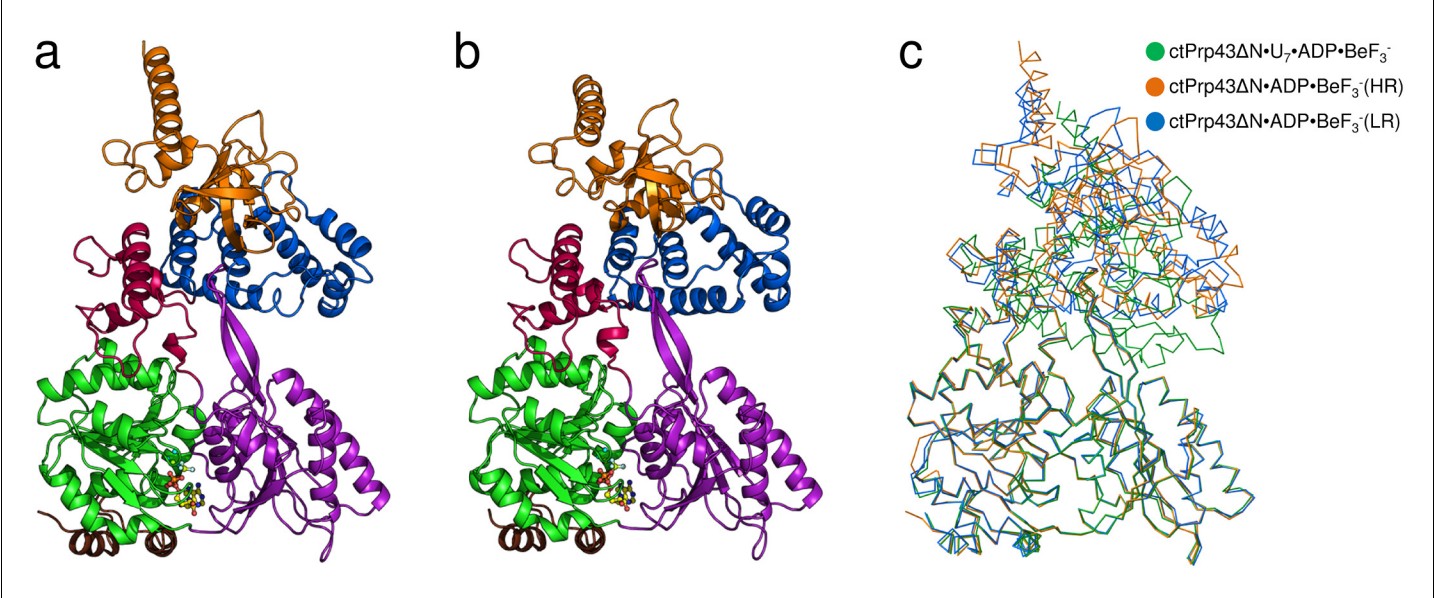

**Figure 2.** Structures of Prp43 with the bound ATP-analog ADP•BeF$_3^-$ in different crystal forms. (**a**) Overall structure of ctPrp43ΔN•ADP•BeF$_3^-$ at high and (**b**) low resolution. ctPrp43ΔN•ADP•BeF$_3^-$(HR) crystalized in the orthorhombic space group P2$_1$2$_1$2$_1$ and ctPrp43ΔN•ADP•BeF$_3^-$(LR) in the hexagonal space group P6$_5$. Domains are colored according to *Figure 1a*. ADP•BeF$_3^-$ is shown at the binding cleft between the two RecA-like domains in ball-and-stick mode. (**c**) Superposition of ctPrp43ΔN•ADP•BeF$_3^-$(HR) (orange) and ctPrp43ΔN•ADP•BeF$_3^-$(LR) (blue) with ctPrp43ΔN•U$_7$•ADP•BeF$_3^-$ (green). Structures were superimposed *via* their helicase core (RecA1 and RecA2 domains) and are shown in ribbon representation.

The following figure supplement is available for figure 2:

**Figure supplement 1.** The main chain of the Hook-Turn (RF motif) is present in two alternative conformations.

values are equal to 17.0 Å (ratchet-like) and 11.7 Å (OB-fold), respectively. The ctPrp43ΔN•ADP•BeF$_3^-$(HR) structure is in a marginally more closed conformation, hence the ctPrp43ΔN•ADP•BeF$_3^-$ complex structures are not completely identical. They represent two slightly different snapshots of Prp43 which both might exist in solution and presumably are favored by the different crystallization conditions or different crystal packing. These two states of ctPrp43ΔN•ADP•BeF$_3^-$ might also give a hint at how Prp43 switches into the closed conformation since in the ctPrp43ΔN•ADP•BeF$_3^-$(HR) complex structure one inter-domain contact is present between the RecA2 (Asp 486) and the ratchet-like domain (Lys 605). These interactions between sequentially distal residues might trigger the switching from the open into the closed conformation after the binding to RNA. The circumstance that both structures of the ctPrp43ΔN•ADP•BeF$_3^-$ complex are in a similar, open conformation strongly suggests the possibility that the complex structures presented reflect the main conformation of Prp43 in an active state in solution. Owing to the fact that Prp43•ADP•BeF$_3^-$ crystallized in different space groups and in divergent crystallization conditions, the open conformation *in crystallo* is highly unlikely to be a crystallization artefact. This assumption is additionally supported by the low number of crystal contacts for the ctPrp43ΔN•ADP•BeF$_3^-$(HR) (355.3 Å$^2$ of buried surface/1.1% of the total surface) and the ctPrp43ΔN•ADP•BeF$_3^-$(LR) (1524.3 Å$^2$/4.7%) complex structure.

To further strengthen the hypothesis that the opening of the tunnel by the displacement of the C-terminal domains is crucial for the helicase function of Prp43, we designed a mutant of Prp43 which allows us to trap the closed conformation by the introduction of an internal disulfide bond (ctPrp43-IDSB). For this purpose, one cysteine was introduced into the RecA1 domain and another one into the ratchet-like domain at exposed positions to maximize the number of formed disulfide bonds (*Figure 4—figure supplement 1*). The functional impact of these mutations was analyzed by an ATPase assay, a fluorescence-based helicase assay and an intron-lariat spliceosome (ILS) disassembly assay. First of all, the percentage of formed disulfide bonds was experimentally estimated.

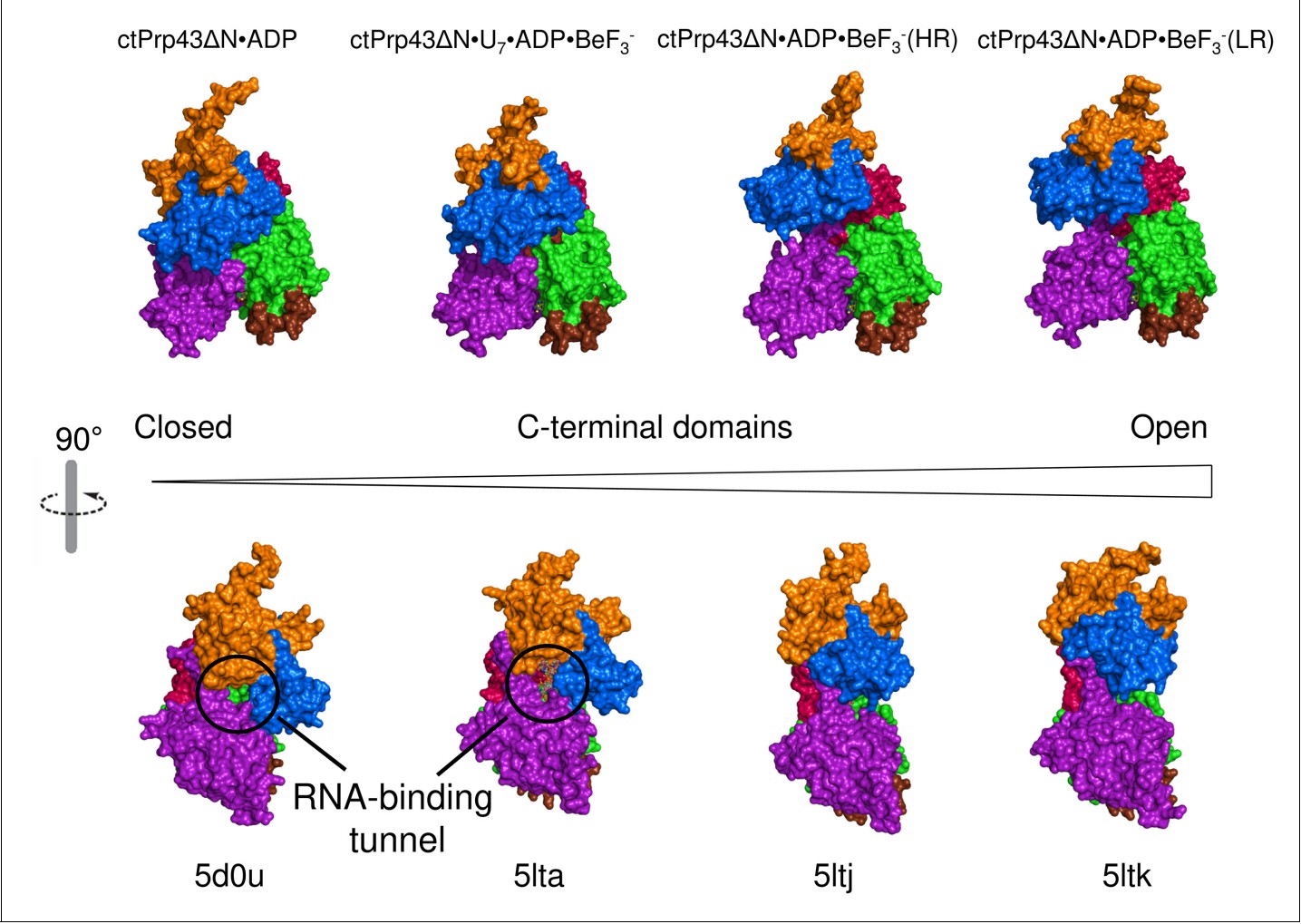

**Figure 3.** Conformational changes between the ctPrp43ΔN•ADP complex structure (5d0u) (*Tauchert et al., 2016*), ctPrp43ΔN•U$_7$•ADP•BeF$_3^-$ (5lta), ctPrp43ΔN•ADP•BeF$_3^-$(HR) (5ltj) and ctPrp43ΔN•ADP•BeF$_3^-$(LR) (5ltk). Structures were superposed *via* their RecA1 domains. In the top panel the back view is presented (rotated by 180° with respect to *Figure 1b*, *Figure 2a* and *Figure 2b*) and in the bottom panel the side view (90° rotation around a vertical axis). Structures are ordered according to the degree of C-terminal displacement and opening of the RNA-binding tunnel. The location of the RNA-binding tunnel which is present in ctPrp43ΔN•ADP and ctPrp43ΔN•U$_7$•ADP•BeF$_3^-$ is indicated.

The wild-type protein contains nine cysteines, the ctPrp43-IDSB mutant two additional ones. *Via* Ellman reaction, the number of cysteines was determined to be 8.5 ± 0.1 (ctPrp43) and 9.3 ± 0.1 (ctPrp43-IDSB), respectively. This allows us to conclude that the majority of ctPrp43-IDSB exhibits the internal disulfide bridge because oxidized cysteines cannot be detected by this method and only the nine cysteines, which are also present in the wild-type protein, were determined.

## Prp43 trapped in the closed conformation is impaired in its helicase activity

Having determined that ctPrp43-IDSB is mainly present in the oxidized and therefore closed state, this mutant was subsequently analyzed concerning its helicase activity using a synthetic dsRNA with a 3'ssRNA overhang (*Figure 4a* and *Table 2*). Since wild-type Prp43 exhibited a very low activity (0.54 nM of unwound RNA/min) in the absence of any stimulatory G-patch (GP) protein, we tested its activity in the presence of the ctNtr1-GP (1.98 nM/min) and the ctPfa1-GP (60.03 nM/min). Since the ctPfa1-GP could increase the Prp43 activity to a much higher level, the ctPfa1-GP was used for the *in vitro* helicase activity analysis of all Prp43 mutants. ctPrp43-IDSB exhibited an activity of 12.18

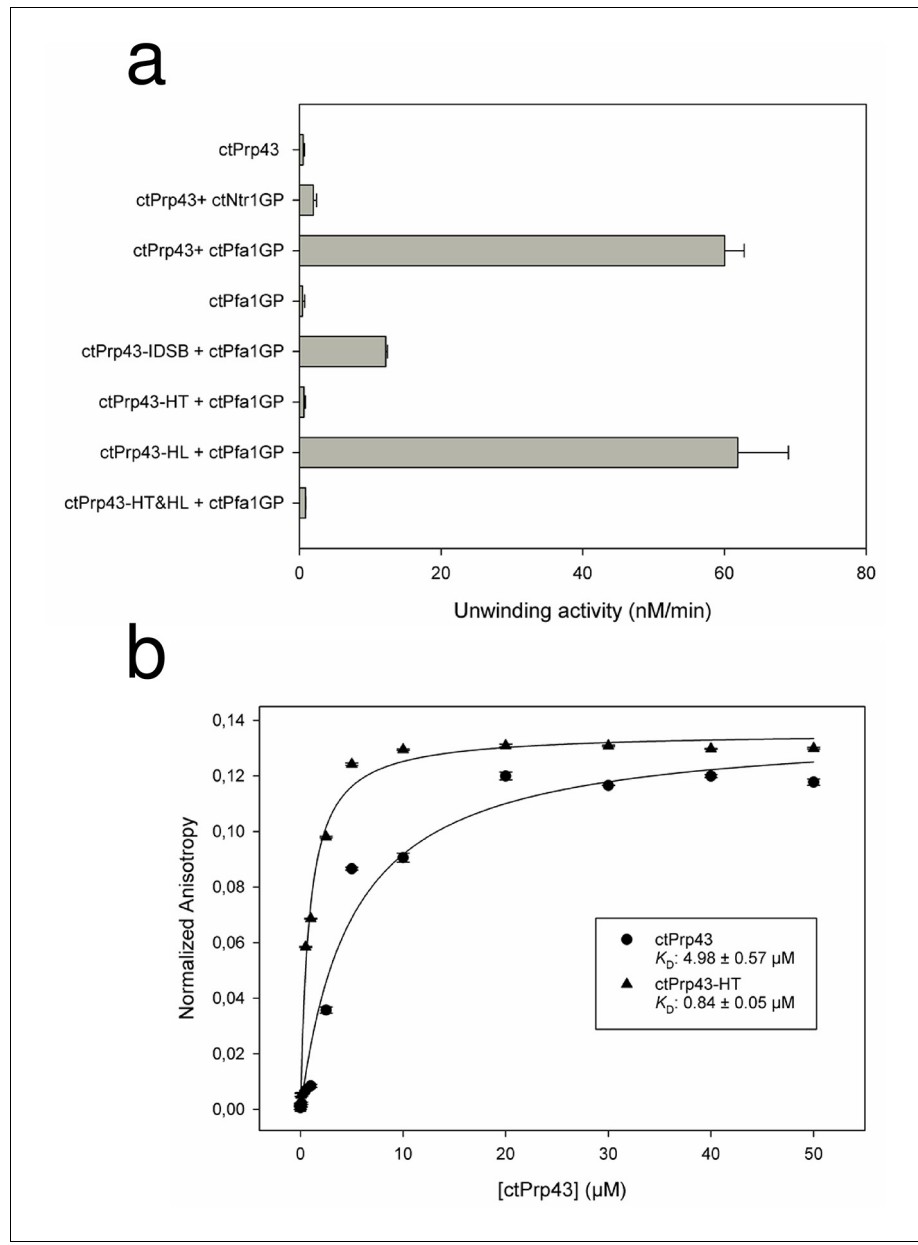

**Figure 4.** Helicase activity and RNA-binding assays of ctPrp43 and mutants. (**a**) The maximal unwinding velocity (nM/min) for a dsRNA with a 3' overhang is shown. (**b**) RNA binding of 5'—6FAM-U$_{16}$-RNA by ctPrp43 and ctPrp43-HT was determined *via* fluorescence anisotropy measurements. Error bars indicate the standard deviation from three independent measurements for **a** and **b**.

The following figure supplements are available for figure 4:

**Figure supplement 1.** Position of the two introduced cysteine residues in the ctPrp43-IDSB mutant.

**Figure supplement 2.** ATPase activity of Prp43 and mutants (**a**) without further stimulation, (**b**) in the presence of a G-patch, (**c**) in the presence of a U$_{16}$-RNA and (**d**) in the presence of a G-patch and a U$_{16}$-RNA.

**Figure supplement 3.** Raw data of exemplary helicase activity measurements.

**Table 2.** Helicase activity.

| | nM/min | ± |
|---|---|---|
| ctPrp43 | 0.54 | 0.18 |
| ctPrp43 + ctPfa1-GP | 60.03 | 2.75 |
| ctPrp43 + ctNtr1-GP | 1.98 | 0.45 |
| ctPfa1-GP | 0.46 | 0.28 |
| ctPrp43-IDSB + ctPfa1-GP | 12.18 | 0.23 |
| ctPrp43-HT+ ctPfa1-GP | 0.68 | 0.16 |
| ctPrp43-HL + ctPfa1-GP | 61.87 | 7.14 |
| ctPrp43-HT&HL+ ctPfa1-GP | 0.88 | 0.03 |

nM/min which equals about 1/5th of the wild-type Prp43 activity and thus is clearly impaired concerning its ability to unwind dsRNA. The remaining helicase activity of the mutant is most likely caused by the fraction of ctPrp43-IDSB which does not contain the internal disulfide bond. The activity of ctPrp43 and the ctPrp43-IDSB mutant were also analyzed for a physiological and more complex substrate RNP by using the *S. cerevisiae* ILS disassembly assays (*Figure 5* and *Figure 5—figure supplement 1*). In previous work it was shown that ctPrp43 is capable of functionally replacing scPrp43 in the spliceosome (*Tauchert et al., 2016*). ctPrp43 can release about 60% of intron-lariat RNA from scILS in the presence of ATP and scNtr(1•2) and dissociate the remaining spliceosomal RNP core into the free U6 snRNA, 18S U5 and 20S U2 snRNP (*Figure 5c*). This is close to the 65% efficiency of recombinant scPrp43 (*Figure 5b*). In case of the ctPrp43-IDSB mutant, these assays were performed in the presence and absence of DTT, which can reduce the disulfide bridge between the RecA1 and ratchet-like domain and thus free ctPrp43-IDSB from its locked conformation. Without DTT, the efficiency of ILS disassembly by ctPrp43-IDSB is clearly impaired compared to the wild-type protein since only 10% of all ILSs are dissociated which equals the amount of the negative control (*Figure 5a and d*). In the presence of DTT, the ILS dissociation efficiency of ctPrp43-IDSB can be fully restored to the wild-type activity level of 60% (*Figure 5e*). This clearly hints at the fact that locking Prp43 in the closed conformation distinctly influences its activity as deduced from the crystal structures of ctPrp43ΔN•ADP•BeF$_3^-$ and thus proving the importance of adopting the open conformation for the helicase activity of Prp43 also in the spliceosome (*Video 1*).

To exclude the possibility that the results of the helicase or ILS disassembly assays differ between ctPrp43 and ctPrp43-IDSB due to a reduced ATPase activity of the latter, ATPase activity measurements were performed. For all ctPrp43 variants, the ATPase activity was analyzed for ctPrp43 itself, after the addition of a GP, or U$_{16}$-RNA or in the presence of both. Since this was the first enzymatic characterization of Prp43 from *C. thermophilum*, several general results are worth mentioning (*Table 3* and *Figure 4—figure supplement 2*). The basal ATPase activity of ctPrp43 (1.81 min$^{-1}$) is very similar to its ortholog from *S. cerevisiae* (2.43 min$^{-1}$) (*Christian et al., 2014*). ctPrp43 is stimulated by both, the ctNtr1-GP and the ctPfa1-GP, but the effect by the latter is more pronounced. A clear difference between *C. thermophilum* and *S. cerevisiae* Prp43 is that ctPrp43 does not exhibit RNA-stimulated ATPase activity in the absence of a GP. The ctPrp43 ATPase activity can only be induced by RNA if an additional G-patch protein is present.

The intrinsic ATPase activity of ctPrp43-IDSB (5.04 min$^{-1}$) is similar to the one determined for wild-type ctPrp43. ctPrp43-IDSB is also stimulated by ctPfa1-GP and by U$_{16}$-RNA in the presence of the ctPfa1-GP, but in the contrast to wild-type Prp43 also just by U$_{16}$-RNA. These findings confirm that the difference in the helicase and the ILS disassembly assay between ctPrp43 and ctPrp43-IDSB is solely caused by the impairment of the unwinding capabilities. The RNA-induced stimulation of ctPrp43-IDSB in the trapped conformation might still be feasible due to the fact that a short single-stranded U$_{16}$-RNA without any secondary structures was used for these assays. Binding to the entry or exit site of the RNA-binding tunnel, which does not require an opening of the tunnel, might be sufficient for the RNA-induced stimulation of the ATPase activity (see below). Furthermore, Prp43 in the trapped closed conformation appears to be more prone for the stimulation of the ATPase

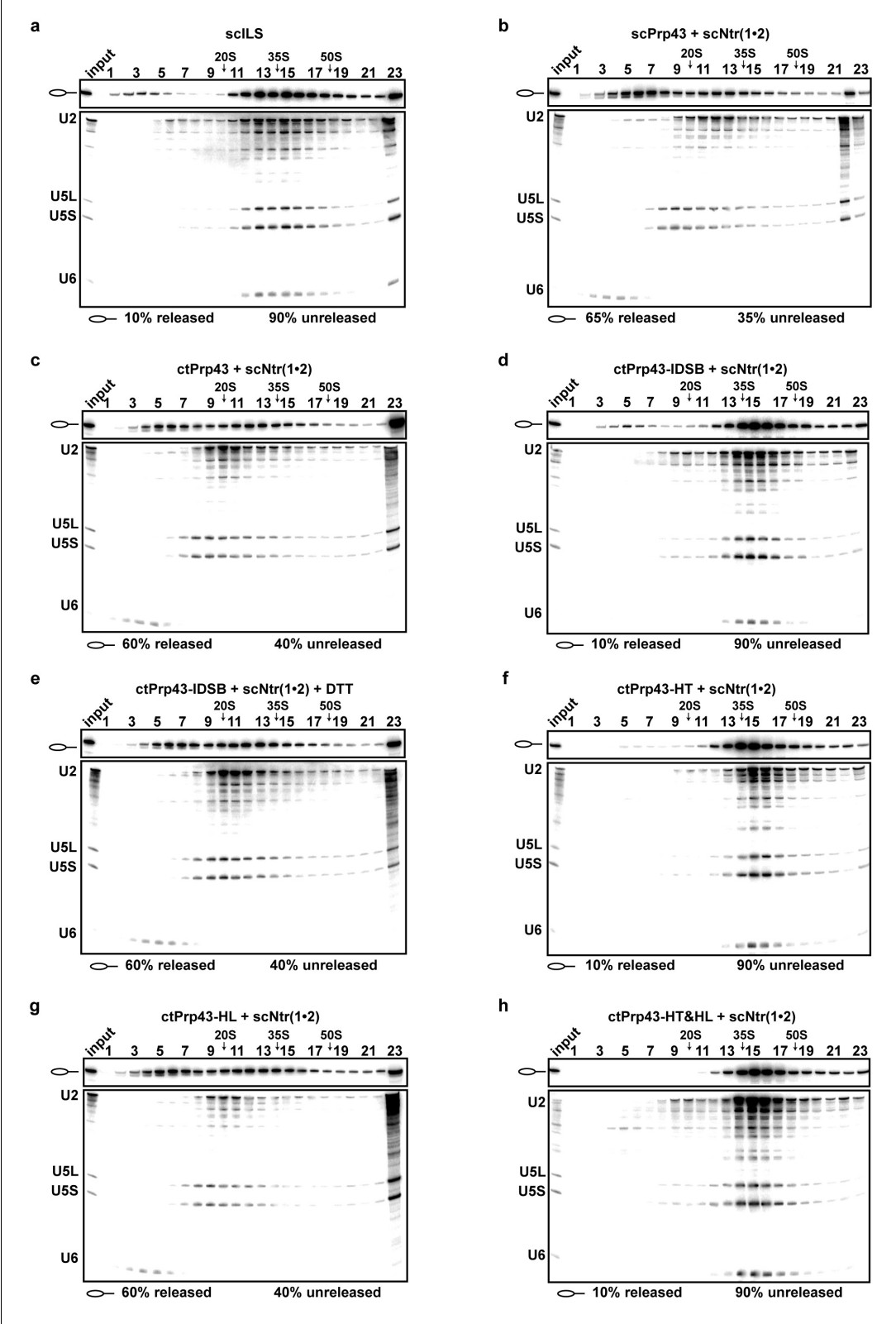

**Figure 5.** Intron-lariat spliceosome (ILS) disassembly assays. 10–30% glycerol gradient sedimentation of purified yeast ILS (scILS) incubated in solution with ATP plus (**a**) no recombinant protein, (**b**) scPrp43 and cofactors scNtr(1•2), (**c**) ctPrp43 and scNtr(1•2), (**d**) ctPrp43-IDSB and scNtr(1•2), (**e**) ctPrp43-IDSB, scNtr(1•2) and 0.5 mM DTT, (**f**) ctPrp43-HT and scNtr(1•2), (**g**) ctPrp43-HL and scNtr(1•2), (**h**) ctPrp43-HT&HL and scNtr(1•2). U2, U5 and U6 snRNAs were visualized by Northern blotting followed by autoradiography. RNA identities are indicated on the left. Quantifications were performed with
*Figure 5 continued on next page*

*Figure 5 continued*

ImageQuant software (Molecular Dynamics). Numbers represent the percentage of intron-lariat RNA released in the top fractions (sum of fractions 1–11) or associated with the ILS (unreleased, sum of fractions 12–23) relative to the intron-lariat RNA distributed in all 23 fractions, the sum of which was set to 100%.

The following figure supplement is available for figure 5:

**Figure supplement 1.** Isolation of intron-lariat spliceosomes (ILSs).

activity by RNA since the ctPrp43-IDSB mutant (24.99 min$^{-1}$) is stimulated by U$_{16}$-RNA while the wild-type protein is not (2.08 min$^{-1}$). It is conceivable that the GP increases the number of Prp43 molecules in a closed conformation since in the presence of a GP also the wild-type protein is stimulated by RNA.

Having shown the importance of the opening of Prp43's C-terminal domains for its unwinding capability, we asked how Prp43 couples ATP hydrolysis and RNA translocation, leading to its helicase activity. Since high-resolution structures of the pre- and post-catalytic state of a DEAH-box helicase are both now available, conformational rearrangements of Prp43 in the course of nucleotide hydrolysis were analyzed.

## Conformational rearrangements at the helicase core

To investigate the effect of ATP hydrolysis on the Prp43 conformation, the structures of the ADP and the ADP•BeF$_3^-$ bound state were compared. Since the conformations of the RecA-like domains of the three structures ctPrp43$\Delta$N•U$_7$•ADP•BeF$_3^-$, ctPrp43$\Delta$N•ADP•BeF$_3^-$(HR) and ctPrp43$\Delta$N•ADP•BeF$_3^-$(LR) are very similar, the analysis of the conformational switches was restricted to the comparison of ctPrp43$\Delta$N•ADP•BeF$_3^-$(HR) with the previously published ctPrp43$\Delta$N•ADP structure.

The individual RecA-like domains of ctPrp43$\Delta$N•ADP and ctPrp43$\Delta$N•ADP•BeF$_3^-$(HR) superimpose well with r.m.s.d. values of 0.36 Å (139 C$_\alpha$) for the RecA1 domains and 0.44 Å (142 C$_\alpha$) for the RecA2 domains. Superimposing the helicase cores consisting of both RecA-like domains shows that the RecA1 and RecA2 domains are rotated by ca. 4° and ca. 14°, respectively. There is also a slight translational displacement of the RecA2 domain by a 2.7 Å shift of its center of mass with regard to the RecA1 domain. Depending on the ADP/ATP state, conformational changes of several of the conserved SF2 helicase motifs, most strikingly of those in the RecA2 domain are observed (*Figure 6*). The conformations of the motifs I, Ia, Ib and II, which are all located in the RecA1 domain, are similar between the ATP-bound and the ADP-bound state of Prp43. This ATP-bound state, mimicked by the bound ADP•BeF$_3^-$, also allows to deduce how Prp43 does hydrolyze ATP. The Glu 219 of the eponymous DEAH-motif (motif II) binds a water molecule (H$_2$O 388) and positions it in close spatial proximity to the BeF$_3^-$, which corresponds to the ATP γ-phosphate. Presumably, this H$_2$O 388 performs the nucleophilic attack on the phosphorus atom of the γ-phosphate since it is in an almost perfect orientation for an S$_N$2 substitution mechanism at 180° to the leaving group. In the ADP•BeF$_3^-$ complex structure, this water molecule is additionally bound *via* the interactions with Gln 428 and Arg 435 from motif VI. The region containing the motif VI undergoes one of the most pronounced conformational rearrangements of the helicase core and it is shifted in the ADP-bound

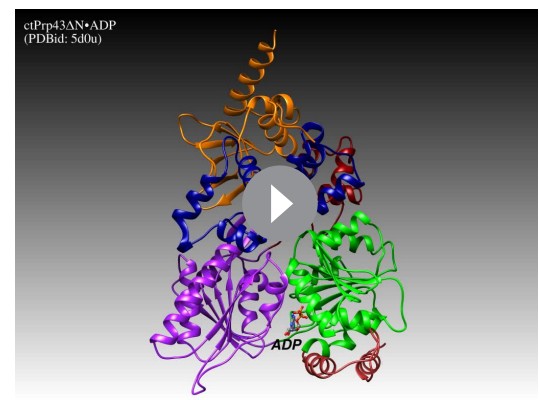

ctPrp43ΔN•ADP
(PDBid: 5d0u)

ADP

**Video 1.** Prp43 adopts an open conformation after ATP binding and switches into the closed conformation after binding to RNA. This morphing movie between the ADP-bound state of Prp43 (PDBid: 5d0u), the two ADP•BeF$_3^-$ bound states (PDBids: 5ltk and 5ltj) and the ctPrp43ΔN•U$_7$•ADP•BeF$_3^-$ complex structure (PDBid: 5lta) illustrates the RNA-binding mode of Prp43.

**Table 3.** ATPase activity.

| | $k_{cat}$ (min$^{-1}$) | ± | $K_M$ (µM) | ± |
|---|---|---|---|---|
| ctPrp43 | 1.81 | 0.03 | 47.23 | 3.40 |
| ctPrp43 + U$_{16}$-RNA | 2.08 | 0.05 | 3.75 | 0.68 |
| ctPrp43 + ctNtr1-GP | 4.16 | 0.04 | 5.49 | 0.32 |
| ctPrp43 + ctNtr1-GP + U$_{16}$-RNA | 16.28 | 0.09 | 4.87 | 0.19 |
| ctPrp43 + ctPfa1-GP | 36.28 | 0.36 | 8.90 | 0.58 |
| ctPrp43 + ctPfa1-GP + U$_{16}$-RNA | 372.20 | 2.91 | 26.11 | 1.21 |
| ctPrp43-IDSB | 5.04 | 0.12 | 16.23 | 2.38 |
| ctPrp43-IDSB + ctPfa1-GP | 199.35 | 1.59 | 29.04 | 1.35 |
| ctPrp43-IDSB + U$_{16}$-RNA | 24.99 | 0.37 | 3.42 | 0.38 |
| ctPrp43-IDSB + ctPfa1-GP + U$_{16}$-RNA | 609.38 | 6.15 | 88.14 | 3.50 |
| ctPrp43-HT | 13.21 | 0.18 | 91.23 | 5.89 |
| ctPrp43-HT + ctPfa1-GP | 60.95 | 0.74 | 13.29 | 1.03 |
| ctPrp43-HT + U$_{16}$-RNA | 10.58 | 0.34 | 53.92 | 8.27 |
| ctPrp43-HT + ctPfa1-GP + U$_{16}$-RNA | 98.61 | 1.84 | 13.19 | 1.52 |
| ctPrp43-HL | 6.19 | 0.05 | 14.31 | 0.71 |
| ctPrp43-HL + ctPfa1-GP | 86.50 | 0.84 | 14.38 | 0.89 |
| ctPrp43-HL + U$_{16}$-RNA | 6.16 | 0.06 | 5.38 | 0.35 |
| ctPrp43-HL + ctPfa1-GP + U$_{16}$-RNA | 533.55 | 9.53 | 49.49 | 3.86 |
| ctPrp43-HT&HL | 7.61 | 0.11 | 16.72 | 1.49 |
| ctPrp43-HT&HL + ctPfa1-GP | 113.18 | 1.03 | 25.17 | 1.37 |
| ctPrp43-HT&HL + U$_{16}$-RNA | 5.53 | 0.07 | 5.59 | 0.51 |
| ctPrp43-HT&HL + ctPfa1-GP + U$_{16}$-RNA | 82.28 | 1.31 | 11.88 | 1.18 |

state 5.1 Å apart from the bound nucleotide. In the ATP-bound state, also interactions between Arg 432 and Arg 435 from motif VI and the phosphate(-mimic) groups are detectable. In the ADP-bound state, only Arg 435 of motif VI interacts with the 3'OH of the ribose. Another noticeable rearrangement can be observed for Phe 360 which is not part of a classical SF2 helicase motif. In the ADP-bound state, the adenine moiety interacts with Phe 360 *via* π-electron stacking and by cation-π-interactions with Arg 162. In the ATP-bound state, Phe 360 is shifted 5.4 Å apart from the base which is now solely stabilized by Arg 162. The rearrangements of the other conserved motifs are less distinct but still notably. In the ADP-bound state, the motifs IV and V are shifted by 3.3 Å and 2.9 Å, respectively, apart from the C-terminal domains.

In the course of NTP hydrolysis, the loop harbouring the conserved motif III, which is known to couple ATPase and unwinding activity (*Schwer and Meszaros, 2000*; *Banroques et al., 2010*; *Fitzgerald et al., 2016*), also undergoes a pronounced rearrangement. The transition of this motif might be required to induce the global conformational rearrangements leading to the rotation of the RecA domains. The rotation leads to a movement of a β-turn in the RecA1 domain and of a β-hairpin in the RecA2 domain, which both are directly located at the RNA-binding tunnel and thus might be involved in RNA translocation (*Figure 7a*, *Video 2*). The β-turn in the RecA1 domain consists of the amino acids Arg 180 and Phe 181, and the β-hairpin of the RecA2 domain is composed of the amino acids Tyr 348, Gly 349 and Thr 350. Interestingly, this second loop has previously been identified to be crucial for the helicase activity of the DExH-box helicase MLE (*Prabu et al., 2015*). In this study by Prapu and co-workers, a triple-mutant of the MLE was generated in which all residues located in this loop were mutated to glycine residues. This loop was termed Hook-Loop as it hooks between the bases of the bound RNA. It was assumed that in the course of NTP hydrolysis the movement of this loop causes the translocation of the bound RNA. Replacing the wild-type Hook-Loop sequence by GGG led to a complete loss of the helicase activity of MLE. Notably, in contrast

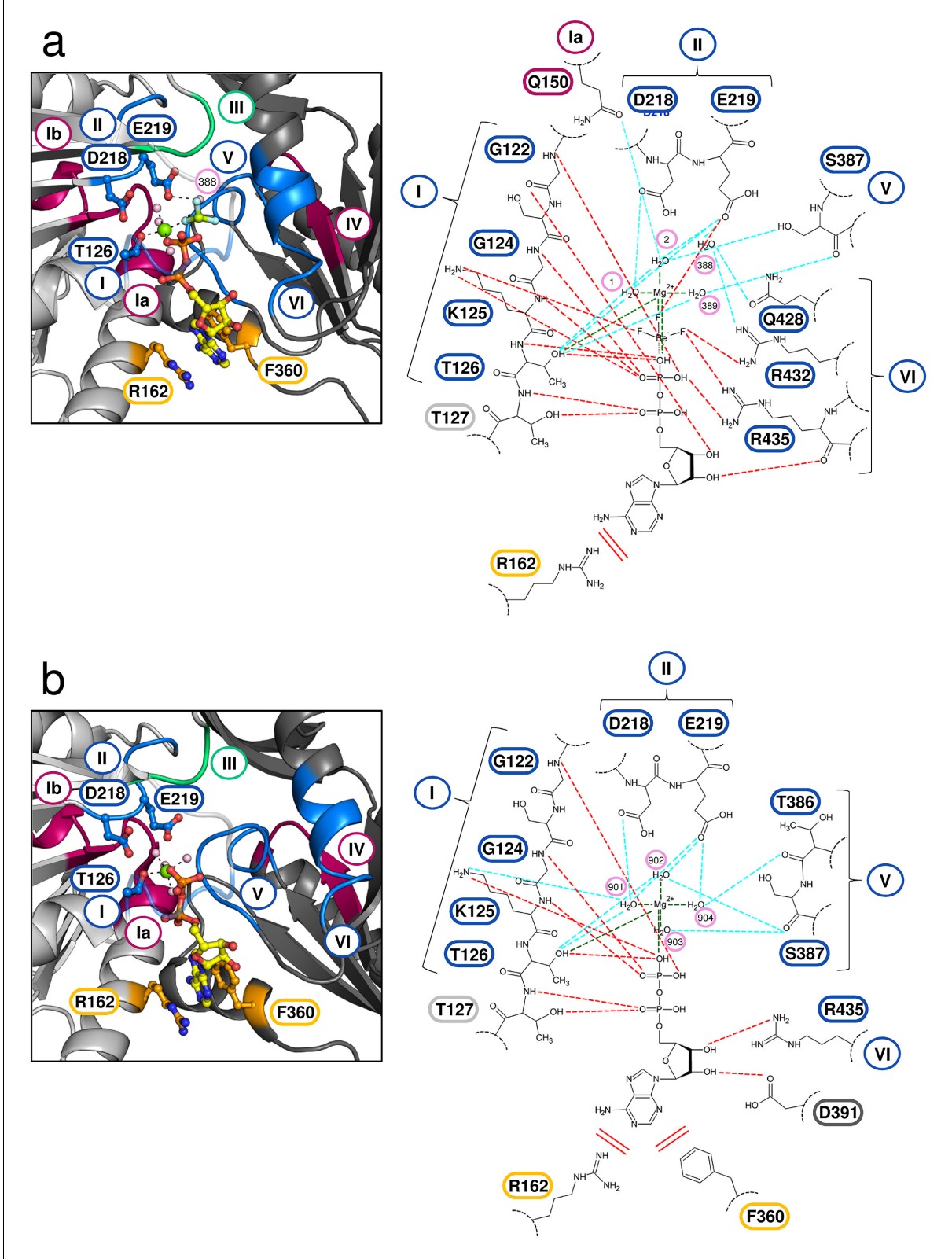

**Figure 6.** Active site of Prp43 in the ATP- and ADP-bound state. (a, Left panel) Active site of Prp43 with the bound ATP-analog ADP•BeF$_3^-$ as present in ctPrp43ΔN•ADP•BeF$_3^-$(HR) (PDBid: 5ltj). The RecA1 domain is colored in light gray, the RecA2 domain in dark gray, carbon atoms of the ADP in yellow, oxygen in red, nitrogen in blue, phosphorus in orange, beryllium in chartreuse, fluoride in light blue, magnesium in light green and water molecules in pale pink. Nucleotide-interacting motifs are shown in blue, nucleic acid-binding motifs in ruby and motif III, which couples ATP hydrolysis to RNA

*Figure 6 continued on next page*

*Figure 6 continued*

unwinding, in green. Residues, which are involved in base-stacking with the adenine moiety, are presented in light orange. Residues from motif I, which are involved in $Mg^{2+}$ coordination (Thr 126), and from motif II, which coordinate water molecules at the active site (Asp 218 and Glu 219), are presented as sticks. The water molecule 388 is perfectly positioned for the nucleophilic attack on the γ-phosphate and thus its hydrolysis. The conserved SF2 helicase motifs I–VI are numbered according to convention. (a, Right panel) Schematic representation of the active site in ctPrp43ΔN•ADP•BeF$_3^-$(HR). Motifs and residues are labeled as introduced in the left panel. Water molecules are numbered according to the PDB entry. Stacking interactions are shown by double lines, polar interactions between ADP•BeF$_3^-$ and Prp43 *via* red dotted lines, interactions between Prp43 and water molecules *via* light blue lines and the coordination of the central $Mg^{2+}$ by dark green lines. (b) Active site of Prp43 with bound ADP (PDBid: 5d0u) (*Tauchert et al., 2016*). (Left and right panel) Labeling and numbering according to **a**. Numerous rearrangements and conformational changes are noticeable between the ATP- and ADP-bound state.

The following figure supplement is available for figure 6:

**Figure supplement 1.** Superposition of the RecA-like domains of the ctPrp43ΔN•ADP•BeF$_3^-$ (HR) structure with chain A of the MLE•U$_{10}$•ADP•AlF$_4^-$ complex (PDB 5aor) (*Prabu et al., 2015*).

to MLE the corresponding residues of the Prp43 Hook-Loop do not contact the bound RNA. In order to analyze the impact of both β-hairpin loops on RNA unwinding, several mutants of Prp43 were generated. In the Hook-Turn mutant (ctPrp43-HT), the amino acids 180–181 were mutated to GG and in the Hook-Loop mutant (ctPrp43-HL) the residues 348 and 350 were replaced by glycines. In addition to this, a double mutant in which both, the Hook-Turn and the Hook-Loop, contain the glycine substitutions (ctPrp43-HT&HL), was also analyzed.

## Prp43 translocates RNA *via* its Hook-Turn

The Hook-Turn and Hook-Loop mutants were tested for their dsRNA unwinding capabilities. As shown in *Figure 4a*, the ctPrp43-HT (0.68 nM/min) and the ctPrp43-HT&HL (0.88 nM/min) mutants exhibit a strongly impaired RNA unwinding activity compared with wild-type ctPrp43 (60.03 nM/min) which corresponds to an almost complete loss of function as seen for the negative control (0.46 nM/min). Intriguingly, the ctPrp43-HL (61.87 nM/min) mutant shows an activity being almost equal to the wild-type protein. This high helicase activity of the ctPrp43-HL mutant is in clear contrast to the results for MLE and thus indicates a different RNA translocation mechanism between the two subfamilies of DEAH-box and DExH-box helicases. This difference in the translocation mechanism is consistent with the ctPrp43ΔN•U$_7$•ADP•BeF$_3^-$ complex structure. An interaction between the Hook-Loop and the RNA, as seen in MLE, is unlikely in Prp43 because the Hook-Loop residues point at the opposite direction and do not contact the RNA (*Figure 7b*). In contrast, Arg 180 and Phe 181 of the Hook-Turn directly point at the RNA, and after ATP-hydrolysis these residues are moving towards the RNA (*Figure 7b*) and presumably hook in between the bases. The Hook-Turn exhibits some flexibility since its main chain is present in two alternative conformations in the ctPrp43ΔN•ADP•BeF$_3^-$(HR) complex structure. According to this structure, this β-turn is present as type I (47%) and type II (53%) (*Figure 2—figure supplement 1*).

The assumption that the Hook-Turn is the key player of RNA translocation in Prp43 is additionally strengthened by the results of the ILS disassembly assay (*Figure 5f, g and h*). The ctPrp43-HL mutant dissociates ILSs at the wild-type protein level ~60% (*Figure 5c and g*), while the activity of both ctPrp43-HT and ctPrp43-HT&HL is abolished and is similar to the background level of scILS at ~10% (*Figure 5a, f and h*). These data confirmed that also in the spliceosome only the Hook-Turn residues are crucial for the function of Prp43, while the Hook-Loop residues are not required.

In order to address the possibility that the reduced helicase activity of the HT mutant is caused by a reduced affinity for RNA, the RNA binding was measured by means of fluorescence anisotropy revealing that this mutant is able to bind U$_{16}$-RNA even with a higher affinity than the wild-type protein ($K_D$ 0.84 ± 0.05 μM vs. 4.98 ± 0.57 μM) (*Figure 4b*).

Furthermore, the loss of helicase activity of the HT mutant is not caused by a loss of ATPase activity since all three mutants (ctPrp43-HT: 13.21 min$^{-1}$, ctPrp43-HL: 6.19 min$^{-1}$, ctPrp43-HT&HL: 7.61 min$^{-1}$) exhibit a level of intrinsic ATPase activity comparable to wild-type Prp43 (*Table 3*). The increased activity of the HT mutant is presumably caused by the fact that the arginine from the Hook-Turn is involved in hydrogen-bonding which has to be disrupted in the course of ATP

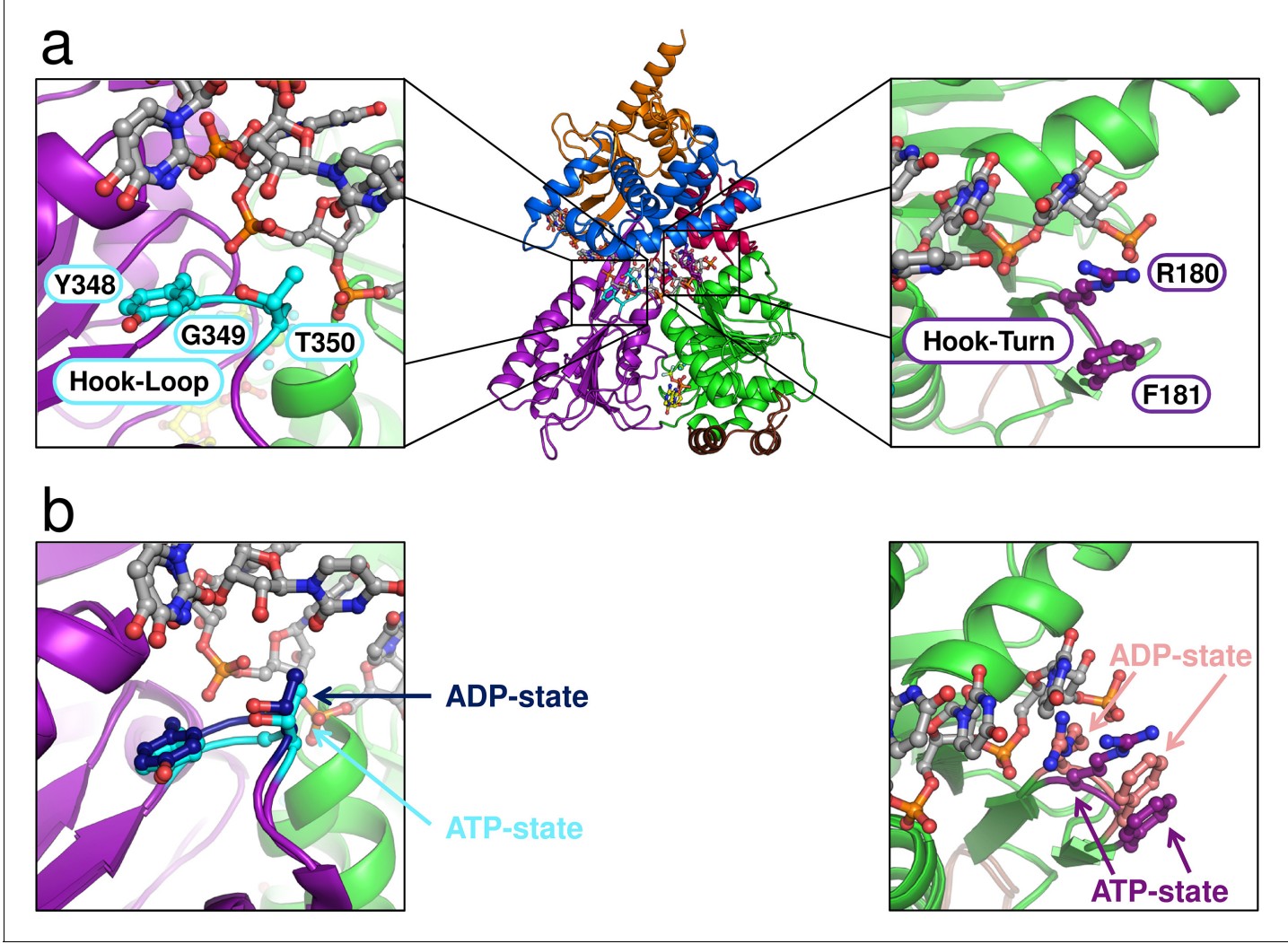

**Figure 7.** Position of the Hook-Turn and Hook-Loop in ctPrp43. (**a**) The localization of the Hook-Turn in the RecA1 domain and of the Hook-Loop in the RecA2 domain in the ctPrp43ΔN•U$_7$•ADP•BeF$_3^-$ complex structure is shown. Domains are colored according to *Figure 1a*. (**b**) Superpositions of the RecA1 and RecA2 domains of the ctPrp43ΔN•U$_7$•ADP•BeF$_3^-$ and the ctPrp43ΔN•ADP (PDB 5d0u) complexes for the Hook-Turn and Hook-Loop, respectively. The superpositions indicate that the Hook-Loop remains in a highly similar conformation after ATP hydrolysis in contrast to the Hook-Turn which is shifted towards the RNA in the ADP-bound state.

The following figure supplement is available for figure 7:

**Figure supplement 1.** Partial sequence alignment of Prp43 from *C. thermophilum* to all DEAH-box RNA helicases from *S. cerevisiae* and of MLE from *D. melanogaster* to all yeast DExH-box RNA helicases.

hydrolysis. In the corresponding Hook-Turn mutants these contacts are not present due to the substitution by glycines, and therefore less energy is consumed by the conformational rearrangements. Furthermore, all of the three Hook-Turn or Hook-Loop mutants are stimulated by G-patch proteins, but it is remarkable that the ctPrp43-HT and ctPrp43-HT&HL mutants are not additionally stimulated by U$_{16}$-RNA in contrast to the ctPrp43-HL mutant. The differences between the ATPase activity of ctPrp43-HT and ctPrp43-HT&HL in the presence of the G-patch protein and of the G-patch protein with RNA are not significant. The lack of RNA-induced stimulation for these two constructs might hint at the fact that the ATPase activity stimulation of Prp43 by RNA occurs at the interior tunnel *via* the interaction with the Hook-Turn.

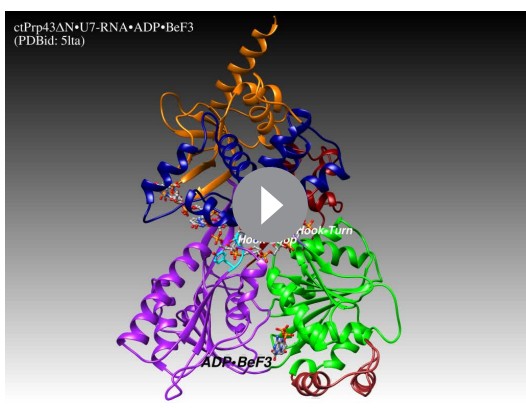

ctPrp43ΔN•U7-RNA•ADP•BeF3
(PDBid: 5lta)

**Video 2.** Local conformational rearrangements at the active site induce global conformational changes in Prp43 which are coupled with the unwinding activity. The morphing between the ctPrp43ΔN•U$_7$•ADP•BeF$_3^-$ (PDBid: 5lta) complex structure and ctPrp43ΔN•ADP (PDBid: 5d0u) reveals conformational rearrangements in the course of ATP hydrolysis, mainly of the Hook-Turn, which is in close spatial proximity to the RNA.

## Discussion

Prp43 was known to bind RNA in a sequence-independent manner which is explained by the ctPrp43ΔN•U$_7$•ADP•BeF$_3^-$ complex structure revealing the almost entire absence of interactions between the bases of the RNA and Prp43. The observed $K_D$ value in the µM range suggests that the RNA binding of Prp43 is transient, which is in line with the low number of hydrogen bonds primarily formed by four out of the seven nucleotides, and the high $B$-factors for U1-U3 and their alternative conformations. However, this unselective and transient binding mode of RNA by Prp43 is perfectly in line with its molecular function. Prp43 helicase activity is required in several distinct cellular processes. Target sites in rRNA maturation and pre-mRNA splicing are very different and thus selective RNA binding would not be compatible with Prp43's mode of function. Furthermore, a tight binding of substrate RNA would be undesirable since it would decrease the efficiency of Prp43 and prolong the required time for the release of substrate RNA.

The comparison of the RNA-binding mode of Prp43 with that of MLE, a DExH-box protein from *D. melanogaster* that was previously crystallized with a U$_{10}$-RNA and the transition state analog ADP•AlF$_4^-$ (*Prabu et al., 2015*), reveals several differences but also striking similarities. Despite the fact that the architecture of the conserved domains is almost identical and the relative position of the bound RNA in the helicase is similar, MLE binds poly-uridine RNA in a sequence-dependent manner. Numerous bases are specifically recognized by MLE and a tighter and much more elaborated interaction network between MLE and the RNA is present. Nonetheless, there are also noticeable similarities concerning the RNA interactions. In Prp43, four main-chain amide interaction interactions between residues of the RecA1 or RecA2 domains and the RNA are present, one with each of the tightly bound nucleotides (U4-U7). Glu 316 interacts with U4, Gly 349 with U5, Arg 153 with U6 and Arg 180 with U7 (*Figure 1e*). There are also two threonine residues of conserved motifs that interact with the RNA. Thr 195 from motif Ib and Thr 381 from motif V are involved in hydrogen bonding *via* their sides chains with the phosphates of the nucleotides U7 and U5, respectively. Interestingly, these main-chain amide interactions are completely conserved in MLE. Despite the fact that the residues themselves are not conserved in MLE, the position at which the interaction takes place is virtually identical (*Figure 1—figure supplement 4*). The corresponding interacting residues and nucleotides in MLE are Trp 663-U6, Ser 692-U7, Arg 443-U8 and Arg 470-U9. Furthermore, the interactions with the conserved threonine residues are also present in MLE (Thr 486-U9 and Thr 717-U7). Moreover, there are three additional side chain interactions which are conserved between Prp43 and MLE. The interaction of Arg 201 (motif Ib) with the ribose of the U7, the hydrogen bonding of Asn 382 with the ribose of U4 and the interaction of Lys 403 with the phosphate moiety of U4 in Prp43 have identical counterparts in MLE. These counterparts are Arg 492-U9, Asn 718-U6 and Arg 739-U6.

Having unraveled the conserved interactions with RNA between the two members of the DEAH/RHA subfamilies, the question arose whether these interactions are present also in the related family of viral NS3/NPH-II helicases. Members of the NS3/NPH-II family also contain a helicase core composed of two RecA-like domains with the protruding β-hairpin in the RecA2 domain. In the NS3/NPHII helicase family there is only one C-terminal domain present, which is not homologous to any C-terminal domain of the DEAH/RHA helicases, and no RNA-binding tunnel is formed. Owing to the fact that numerous NS3 helicases are encoded by human pathogenic viruses, such as hepatitis C, Zika or yellow fever, these helicases have been subject of intensive studies and various crystal structures are available. The comparison with Prp43 was restricted to the NS3 hepatitis C virus (HCV)

helicase, as NS3 HCV binds RNA also in a sequence-independent manner and a crystal structure with a bound ATP-analog (ADP•BeF$_3^-$) and U$_8$-RNA is available making a direct comparison with Prp43 very feasible (PDBid: 3o8r) (*Appleby et al., 2011*). The comparison between Prp43 and NS3 HCV revealed that the unspecific interactions between the RNA backbone and the main-chain amides as well as the two interactions with the conserved threonines from motif Ib and V are also present in NS3 HCV (*Figure 1—figure supplement 4*). In NS3 HCV, the main-chain amides of the residues Lys 371, Arg 393, Val 232 and Gly 255 interact with the nucleotides U4-U7 and Thr 269 as well as Thr 411 are involved in hydrogen bonding *via* their site chains with U7 and U5, respectively. Hence, this RNA-binding mode appears to be conserved for the sequence-unspecific interaction of the RecA-like domains of several members of the DEAH/RHA and NS3 families. However, there are also differences in the RNA binding between the DEAH/RHA and NS3 families, e.g. a functionally important tryptophan of NS3 members (Trp 501 in NS3 HCV), which stacks on the nucleotide at the 3' end, is not conserved in DEAH-box helicases.

With the set of new Prp43 structures novel insights into the mechanism of RNA loading are provided. Considering the fact that the RNA-binding site of Prp43 is in its interior tunnel and the circumstance that the substrates of Prp43 are highly folded RNA networks, RNA loading to Prp43 is not trivial. The Prp43 target sites in the spliceosome and in the pre-ribosomal subunits are surrounded by complex tertiary structures without single-stranded overhangs. Here, we could demonstrate that in the ATP-bound state Prp43 is capable of adopting an open conformation in which numerous interactions between the RecA-like domains and the C-terminal domains are disrupted converting the RNA-binding tunnel into a groove and thus enabling Prp43 to bind to single-stranded RNA regions within folded RNA stretches (*Figure 3*). This rearrangement of the C-terminal domains of Prp43 with bound ADP•BeF$_3^-$ has not been observed in any other crystal structure of a DEAH- or DExH-box helicase, which demonstrates that Prp43 exhibits a higher degree of flexibility as initially expected from the previously determined crystal structures of Prp43•ADP complexes.

Since structures of both the ATP- and ADP-bound state of Prp43 are now available, an in-depth analysis of conformational rearrangements which occur at the active site is possible. One can assume that these rearrangements almost exclusively occur after the release of the γ-phosphate owing to the fact that the helicase cores of Prp43 with the ATP-analog ADP•BeF$_3^-$ and of MLE (chain A) with the transition state analog ADP-AlF$_4^-$ superpose very well with an r.m.s.d. value of 0.70 Å for 235 C$_\alpha$ (*Figure 6—figure supplement 1*). ATP hydrolysis causes the movement of one β-turn and one loop, located at the RNA-binding site of the RecA1 and RecA2 domain, respectively (*Figure 7a*). The β-turn in the RecA1 domain, the Hook-Turn, was shown to be of crucial importance for Prp43's helicase activity, its capability to disassemble ILSs and also for its RNA-induced ATPase activity. In contrast to this, the Hook-Loop, located in the RecA2 domain, does neither affect the unwinding activity nor the ATPase activity of Prp43. Remarkably, the integrity of this Hook-Loop was known to be essential for the helicase activity of MLE and it was therefore proposed that the Hook-Loop is presumably required in all DExH- but also all DEAH-box helicases for the translocation of RNA (*Prabu et al., 2015*). Unexpectedly, but according to our results, the difference in the mechanism of how DEAH-box and DExH-box helicases do translocate RNA through their interior tunnel seems to be tremendous. For Prp43, exclusively the interaction between the Hook-Turn and the RNA is required for successful translocation of the RNA. The mutation of the Hook-Loop, which is essential in MLE, is effectless in Prp43. Furthermore, the Hook-Turn is not required for RNA binding but it is involved in the RNA-induced stimulation of the ATPase activity. This is supported by the finding that the ctPrp43-IDSB mutant still exhibits RNA-stimulated ATPase activity. For this mutant, RNA binding in the tunnel is unlikely but the stimulation might be caused by the binding of the single stranded RNA only to the tunnel exit as this is formed, among others, by the Hook-Turn.

To further elaborate the difference in the RNA-translocation mechanism between the DEAH- and DExH-box subfamilies the conservation of the Hook-Turn and the Hook-Loop residues in family members from *S. cerevisiae*, ctPrp43 and MLE was analyzed (*Figure 7—figure supplement 1*). The RF motif the Hook-Turn is conserved in all DEAH-box helicases, while in the DExH-box helicases, this motif is only present in MLE. Between DEAH-box and DExH-box helicases, the Hook-Loop is not conserved (Yxx vs. Hxx), but conservation among the individual members is noticeable. These findings are perfectly in line with our interpretation of the activity of the Hook-Turn and Hook-Loop mutants, which clearly hint at the fact that the RNA-translocation mechanism between DEAH- and DExH-box helicases is strikingly different. The Hook-Turn is presumably the key feature of genuine

DEAH-box helicases for RNA translocation through the interior binding tunnel and the Hook-Turn RF motif should be added to the list of conserved DEAH-box helicase motifs. Like for MLE, the structure of the Prp43-RNA complex does not provide any indication for a ratcheting function of the C-terminal helical-bundle domain, which supports the idea of renaming this domain.

## Materials and methods

### Protein purification

ctPrp43 and ctPrp43(61–764) (ctPrp43ΔN) were expressed and purified as previously described (*Tauchert et al., 2016*). The mutants ctPrp43(F181C, N623C), ctPrp43(R180G, F181G), ctPrp43 (Y348G, T350G) and ctPrp43(R180G, F181G, Y348G, T350G) were expressed and purified under identical conditions. These mutants are referred to here as ctPrp43-IDSB, ctPrp43-HT, ctPrp43-HL and ctPrp43-HT&HL. The only deviation from the wild-type protein purification protocol was done for ctPrp43-IDSB. This mutant was subjected to overnight stirring during removal of the GST-tag to increase the formation of intramolecular disulfide bonds.

The homologs of the G-patch proteins Pfa1 and Ntr1 from *C. thermophilum* were identified by BLAST search against the complete *C. thermophilum* genome (*Altschul et al., 1990*). The identified homologs are annotated as 'hypothetical protein CTHT_0048220' and 'hypothetical protein CTHT_0020180' and here termed as ctPfa1 and ctNtr1, respectively. The truncated G-patch constructs ctPfa1(662–742) (ctPfa1-GP) and ctNtr1(242–305) (ctNtr1-GP) were cloned from genomic DNA of *C. thermophilum var. thermophilum* DSM 1495 into pGEX-6P-1 *via* BamHI/SalI and EcoRI/SalI restriction sites. Both constructs were expressed in and purified from *E. coli* BL21(DE3) (Agilent Technologies). Cells were lysed in 50 mM Tris/HCl pH 7.5, 500 mM NaCl and 10 mM EDTA by microfluidization and soluble protein was separated by ultracentrifugation. The N-terminally tagged GST-fusion proteins were loaded onto Glutathione Sepharose 4B (GE Healthcare) in lysis buffer and thoroughly washed with an additional 2 M LiCl. After protein elution in 50 mM Tris/HCl pH 7.5, 500 mM NaCl, 2 mM MgCl$_2$ and 30 mM reduced glutathione, GST-tag removal was realized by incubation with *PreScission Protease* (1:100 (w/w), GE Healthcare). Final polishing was obtained by size exclusion chromatography (Superdex 75, GE Healthcare) and a second GST affinity chromatography step, both in 10 mM Tris/HCl pH 7.5, 200 mM NaCl and 2 mM MgCl$_2$.

### Crystallization and data collection

Complexes of ctPrp43ΔN were crystallized *via* the sitting-drop vapor-diffusion technique by mixing equal volumes of protein solution and crystallization buffer. ctPrp43ΔN•ADP•BeF$_3^-$(LR) was crystallized at a concentration of 5 mg ml$^{-1}$ in 40% (v/v) pentaerythritol propoxylate (5/4 PO/OH), 6% (v/v) ethanol, 16% (v/v) glycerine and 100 mM Tris/HCl pH 8.0 at 20°C. ctPrp43ΔN•ADP•BeF$_3^-$(HR) crystallized at 4.5 mg ml$^{-1}$ in 35% (v/v) 2-methyl-2,4-pentanediol, 1.67% (w/v) PEG 4000 and 100 mM Na HEPES pH 7.0 at 4°C. Crystals of the ctPrp43ΔN•U$_7$-RNA•ADP•BeF$_3^-$ complex were obtained at 20°C by mixing ctPrp43ΔN at 3 mg ml$^{-1}$ with a 2.5-fold molar excess of poly-U$_{16}$-RNA (AXOlabs, Germany) and 2.5% (w/v) PGA-LM, 13% (w/v) PEG 8000 and 100 mM Na Cacodylate pH 6.5. For all crystallization attempts, ADP was used at a tenfold molar excess with respect to ctPrp43ΔN, BeSO$_4$ with a twentyfold and NaF at a sixtyfold excess.

Prior to data collection, crystals of the RNA-complex structure were cryo-protected in 26% (v/v) glycerine. Both ctPrp43ΔN•ADP•BeF$_3^-$ crystals did not require additional cryo-protection. Diffraction data were collected at 100 K on beamline P13, PETRA III, DESY (Hamburg, Germany) and were processed with the *XDS* package (*Kabsch, 2010*). Data collection statistics are tabularized in *Table 1*.

### Structure determination, refinement and analysis

The structure of ctPrp43ΔN•U$_7$-RNA•ADP•BeF$_3^-$ was solved by molecular replacement using *Phaser* as implemented in the CCP4 suite and the ctPrp43ΔN•ADP (PDBid: 5d0u) structure as a search model (*Tauchert et al., 2016*; *McCoy et al., 2007*). It was crucial to split this protein into three search models to obtain a reasonable molecular replacement solution. The RecA1, the RecA2 and the C-terminal domains had to be placed independently from each other. After initial model improvement in *PHENIX* and manual adjustment in *Coot*, the ADP•BeF$_3^-$ and seven nucleotides of the U$_{16}$-RNA, which was present in the crystallization reaction, could be fitted into the difference

density (*Adams et al., 2010*; *Emsley et al., 2010*). Subsequent to additional refinement in *PHENIX* and *Coot*, the final model was refined to $R_{work}$ and $R_{free}$ of 19.70% and 22.97% (for refinement statistics see *Table 1*) and validated with *MolProbity* (*Chen et al., 2010*). 95.98% of all residues are in favored regions of the Ramachandran plot and 0.0% are indicated as outliers.

Both crystal structures of ctPrp43$\Delta$N•ADP•BeF$_3^-$(HR and LR) were determined by means of molecular replacement with *Phaser* using the atomic coordinates of ctPrp43$\Delta$N•U$_7$-RNA•ADP•BeF$_3^-$ as a search model which was split in three distinct parts. The RecA1 with the RecA2 domain, the ratchet-like domain linked to the OB-fold and the WH domain as last item were placed in a stepwise fashion to obtain a successful replacement solution. Subsequent iterative cycles of automated refinement with *PHENIX* and manual model building in *Coot* were carried out before model quality assessment in *MolProbity*. The structure of ctPrp43$\Delta$N•ADP•BeF$_3^-$(HR) was refined to $R_{work}$ and $R_{free}$ values of 17.67% and 19.88% and of ctPrp43$\Delta$N•ADP•BeF$_3^-$(LR) to 18.22% and 21.78% with 97.34% and 96.28% of all residues in the most-favored regions of the Ramachandran plot, respectively. Both structures do not exhibit any Ramachandran outlier. Detailed refinement statistics are summarized in *Table 1*.

Figures were prepared with *PyMOL* (v.1.3; Schrödinger) and *Chimera* (*Pettersen et al., 2004*).

## ATPase activity assays

The ATPase activity of ctPrp43 and all of its mutants was monitored in an NADH-dependent enzyme coupled assay by recording the decrease of NADH absorption at 340 nm as initially described by Agarwal and coworkers (*Agarwal et al., 1978*). All reactions were performed as a set of triplicates in 25 mM Tris/HCl pH 7.5, 150 mM KCl and 3 mM MgCl$_2$ at 25°C on an Ultrospec 2100 pro UV/Vis spectrophotometer (GE Healthcare). Reactions were supplemented with 250 nM NADH, 500 nM phosphoenolpyruvate, 6–8.3 U/ml pyruvate kinase (Sigma-Aldrich, UK) and 9–14 U/ml lactic dehydrogenase (Sigma-Aldrich, UK). ctPrp43 and mutants were used at a concentration range between 0.25–10 µM. G-patch proteins and U$_{16}$-RNA (Sigma-Aldrich, UK) were each added in five-fold excess over ctPrp43. $k_{cat}$ and $K_M$ values were determined according to Michaelis and Menten equation.

## Helicase activity assays

Helicase activity was monitored with a fluorescence-based unwinding assay (*Belon and Frick, 2008*) using a dsRNA substrate with a 3'ssRNA overhang, consisting of 5'-GCG CCU ACG GAG CUG GUG GCG UAG GCG CAA AAA AAA AAA AAA AAA AAA-3' and 5'-(Cy5)-GCG CCU ACG CCA CCA GCU CCG UAG GCG C-(BBQ)-3', as previously established for scPrp43 (*Christian et al., 2014*). In this assay, a decrease of fluorescence is measured which is caused by the quenching of BBQ on Cy5. Quenching only occurs if the dsRNA is disrupted and an internal hairpin is formed within the labeled RNA strand, positioning BBQ and Cy5 in close spatial proximity. Assays were performed in 25 mM Tris/HCl pH 7.5, 150 mM KCl, 3 mM MgCl$_2$ and 1 mM ATP at 20°C and recorded on a Fluoromax III (Horiba Jobin Yvon). ctPrp43 and ctPrp43 mutants were used at a concentration of 250 nM, G-patch proteins at 1.25 µM and the dsRNA (AXOlabs, Germany) at 500 nM. The excitation wavelength was set to 643 nm and the emission was measured at 673 nm with a slit setting of 2.5 nm/2.0 nm. The initial slope ($\Delta$F/s) of each reaction, which is the maximum reaction velocity, was determined in three independent measurements per sample (for example measurements see *Figure 4—figure supplement 3*). To determine the amount of unwound RNA in nM/min, data were normalized with regard to the fluorescence signals at 667 nm for the intact dsRNA and the completely quenched ssRNA, respectively. The mean value of the initial slope (nM/min) for the reactions with ctPrp43 and all ctPrp43 mutants was plotted with the corresponding standard deviation.

## Determination of reduced cysteine residues

The number of reduced cysteines was determined using Ellman's reagent (5,5'-dithiobis(2-nitrobenzoic acid)) (*Ellman, 1959*). Reactions were carried out under denaturing conditions in 8 M urea in the presence of 400 µM Ellman's reagent and 5 µM ctPrp43 or ctPrp43-IDSB. The number of sulfhydryl groups was calculated by measuring the absorption at 412 nm using an Ultrospec 2100 pro UV/Vis spectrophotometer (GE Healthcare) and applying an extinction coefficient of 14290 M$^{-1}$ cm$^{-1}$.

## RNA-binding assay

RNA binding of ctPrp43 and ctPrp43-HT was analyzed *via* fluorescence anisotropy measurements using a Fluoromax VI (Horiba Jobin Yvon). 5' 6-carboxyfluorescein-labeled $U_{16}$-RNA (Sigma-Aldrich, UK) was used at a concentration of 0.1 µM and ctPrp43 or ctPrp43-HT at concentrations between 0.01 µM to 50 µM in the presence of a 100-fold excess of AMPPNP (Jena Bioscience, Germany). RNA-binding assays were performed in 25 mM Tris pH 7.5, 150 mM KCl and 3 mM $MgCl_2$ at 25°C. The excitation wavelength was set to 490 nm and the emission was measured at 517 nm with a slit setting of 5 nm/5 nm. Data points were analyzed as a set of triplicates and for each sample, and the anisotropy was determined as the mean value of ten individual measurements. The measured anisotropy was normalized with respect to the sample with a concentration of 0 µM ctPrp43 or ctPrp43-HT.

## Spliceosome purification and reconstitution

Yeast $B^{act\ \Delta Prp2}$ complexes were assembled in heat-inactivated extracts from the yeast strain *prp2-1* (*Yean and Lin, 1991*) by incubating with Actin7 pre-mRNA containing MS2 aptamers at 23°C for 45 min (*Figure 5—figure supplement 1*). Samples were centrifuged for 10 min at 9000 rpm and loaded onto columns containing 200 µl of amylose matrix equilibrated with GK75 buffer (20 mM HEPES-KOH pH 7.9, 1.5 mM $MgCl_2$, 75 mM KCl, 5% glycerol, 0.01% NP40). The matrix was washed twice with 10 ml GK75 buffer. To obtain ILSs (*Fourmann et al., 2013*), $B^{act\ \Delta Prp2}$ complexes bound to the amylose matrix were supplemented with a 10-fold molar excess of recombinant proteins (Prp2, Spp2, Cwc25, Prp16, Slu7 and Prp18) and the reaction volume was adjusted to 400 µl with GK75 buffer; then 40 µl of 10x 'rescue' solution (200 mM $KPO_4$ (pH 7.4), 10 mM $MgCl_2$, 20 mM ATP, 10% PEG 8000) were added to the reaction that was incubated at 23°C for 45 min. After thorough mixing, the reaction was incubated at 23°C for 45 min. Matrices were subsequently washed 3 times with 10 column volumes of GK75 buffer. Then a 10-fold molar excess of recombinant Prp22 was added and the volume was adjusted to 400 µl with 1x 'rescue' solution prepared in GK75 buffer. After thorough mixing, the reaction was incubated at 23°C for 15 min. The supernatant (containing the released ILS) was collected, and GK75 buffer was added to the matrix to a final volume of 400 µl. After gentle mixing and repeated centrifugation for 1 min at 2000 rpm the supernatant was collected and then loaded onto linear 10–30% (v/v) glycerol gradients containing GK75 buffer. Samples were centrifuged for 16 h at 21,500 rpm in a TH660 rotor (Thermo Scientific) and harvested manually from the top in 23 fractions of 175 µl. Fractions were analyzed by Cherenkov counting in a scintillation counter. Peak fractions containing ILSs were pooled and the glycerol concentration was adjusted to 5% with GK75 buffer without glycerol.

## Spliceosomal disassembly assays

To dismantle purified ILSs (*Fourmann et al., 2016b*), samples were incubated with distinct combinations of a 10-fold molar excess over the spliceosome of recombinant scPrp43, ctPrp43, ctPrp43-IDSB, ctPrp43-HT, ctPrp43-HL, ctPrp43-HT&HL and scNtr(1•2). An additional step of incubation was performed for the variant ctPrp43-IDSB in the presence of 0.5 mM DTT for 5 min at 23°C. The volume was adjusted to 400 µl with $1\times$ 'rescue' solution prepared in GK75 buffer containing ATP and only for the variant ctPrp43-IDSB 0.5 mM DTT was added to the reaction. After thorough mixing, the mixture was incubated at 23°C for 15 min and then subjected to glycerol gradient centrifugation for 2 h at 60,000 rpm in a TH660 rotor and harvested manually from the top in 23 fractions of 175 µl. Each fraction was digested with Proteinase K followed by phenol-chloroform-isoamyl alcohol (PCI) extraction. RNA was precipitated with ethanol, and then analyzed by PAGE on 8% polyacrylamide, 8 M urea gels (PAGE) and visualized by autoradiography or Northern blot analysis.

## Accession codes

Coordinates and structure factors of ctPrp43$\Delta$N•$U_7$•ADP•$BeF_3^-$ (PDBid: 5lta), ctPrp43$\Delta$N•ADP•$BeF_3^-$(HR) (PDBid: 5ltj) and ctPrp43$\Delta$N•ADP•$BeF_3^-$(LR) (PDBid: 5ltk) have been deposited in the Protein Data Bank.

## Acknowledgements

We acknowledge access to the EMBL beamline P13, PETRA III, DESY, Hamburg. This work was supported by grants from the Deutsche Forschungsgemeinschaft (DFG) to RF and RL (SFB 860, TPA2 and TPA1). Furthermore, we are thankful to Dr. Piotr Neumann for support during data collection and structure validation.

## Additional information

### Funding

| Funder | Grant reference number | Author |
|---|---|---|
| Deutsche Forschungsgemeinschaft | SFB860 TPA2 | Ralf Ficner |
| Deutsche Forschungsgemeinschaft | SFB860 TPA1 | Reinhard Lührmann |

The funders had no role in study design, data collection and interpretation, or the decision to submit the work for publication.

### Author contributions

MJT, Conceptualization, Data curation, Formal analysis, Validation, Investigation, Visualization, Writing—original draft, Writing—review and editing; J-BF, Formal analysis, Investigation, Visualization, Writing—review and editing; RL, Supervision, Funding acquisition, Writing—review and editing; RF, Conceptualization, Resources, Data curation, Formal analysis, Supervision, Funding acquisition, Validation, Project administration, Writing—review and editing

### Author ORCIDs

Reinhard Lührmann, http://orcid.org/0000-0002-6403-4432
Ralf Ficner, http://orcid.org/0000-0002-1739-6086

## Additional files

### Major datasets

The following datasets were generated:

| Author(s) | Year | Dataset title | Dataset URL | Database, license, and accessibility information |
|---|---|---|---|---|
| Tauchert MJ, Ficner R | 2016 | Crystal structure of the Prp43-ADP-BeF3-U7-RNA complex | http://www.rcsb.org/pdb/explore/explore.do?structureId=5LTA | Publicly available at the RCSB Protein Data Bank (accession no: 5LTA) |
| Tauchert MJ, Ficner R | 2016 | Crystal structure of the Prp43-ADP-BeF3 complex (in orthorhombic space group) | http://www.rcsb.org/pdb/explore/explore.do?structureId=5LTJ | Publicly available at the RCSB Protein Data Bank (accession no: 5ltj) |
| Tauchert MJ, Ficner R | 2016 | Crystal structure of the Prp43-ADP-BeF3 complex (in hexagonal space group) | http://www.rcsb.org/pdb/explore/explore.do?structureId=5LTK | Publicly available at the RCSB Protein Data Bank (accession no: 5LTK) |

The following previously published datasets were used:

| Author(s) | Year | Dataset title | Dataset URL | Database, license, and accessibility information |
|---|---|---|---|---|
| Tauchert MJ, Ficner R | 2016 | Crystal structure of the RNA-helicase Prp43 from Chaetomium thermophilum bound to ADP | http://www.rcsb.org/pdb/explore/explore.do?structureId=5D0U | Publicly available at the RCSB Protein Data Bank (accession no: 5D0U) |
| Prabu JR, Conti E | 2015 | Structure of MLE RNA ADP AlF4 complex | http://www.rcsb.org/pdb/explore/explore.do?structureId=5AOR | Publicly available at the RCSB Protein Data Bank (accession no: 5AOR) |
| Appleby TC, Somoza JR | 2011 | Visualizing ATP-dependent RNA Translocation by the NS3 Helicase from HCV | http://www.rcsb.org/pdb/explore/explore.do?structureId=3O8R | Publicly available at the RCSB Protein Data Bank (accession no: 3O8R) |

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
