## [Decision Letter]

Thank you for submitting your article "Structural insights into the mechanism of DEAH-box RNA helicases" for consideration by *eLife*. Your article has been favorably evaluated by James Manley (Senior Editor) and three reviewers, one of whom, Timothy W Nilsen (Reviewer #1), is a member of our Board of Reviewing Editors.

The reviewers have discussed the reviews with one another and the Reviewing Editor has drafted this decision to help you prepare a revised submission.

As you will see all were generally positive about the work but also raised some concerns. In particular, it is necessary to test the RNA binding affinity of the hook loop mutants to ensure that the effect on translocation is not simply because of failure of substrate binding. Another point that must be addressed is to put your work in context with work published on DEAH proteins found in flaviviruses. In addition to these points please deal with the other concerns raised as thoroughly as possible.

Reviewer #1:

Ficner and co-workers report the crystal structure of the DEAH RNA helicase Prp43 with bound RNA and an ATP ground state analog. Compared to a previously reported structure of Prpr43 with ADP (but without RNA), the new structure reveals a significant conformational change that is interpreted as facilitating RNA loading and movement of the RNA relative to the protein. The authors further identify a hairpin loop with a characteristic motif that is conserved among DEAH helicases. While the work focuses on structural data, the main structure-inspired claims are supported by biochemical and functional studies in yeast for several mutants.

The new structure provides critical information for understanding helicases of the DEAH family, many of which participate in pre-mRNA splicing. In addition, the work highlights potentially important differences to a recently reported structure of the related MLE helicase bound to RNA. The manuscript thus makes the point that even within the DEAH/RHA family differences exist with respect to the RNA binding mode.

While the data and the overall information mark an important contribution to our understanding of RNA biology, especially pre-mRNA splicing, the presentation leaves room for improvement. The writing is verbose, and the manuscript would probably benefit from tightening of the prose. In addition, the figures do not always convey a clear point and are at times hard to follow (examples include Figure 1, Figure 4, Figure 5). It might be helpful to include more schematic representations of the structural features that are highlighted, and a model at the end that emphasizes the conformational changes for the proposed RNA loading process.

Reviewer #2:

The authors have determined the structure of a DEAH-box helicase, C. thermophilus Prp43, bound to RNA and to ADP-BeF, e.g. in the ground-state of the reaction. Comparing their ground-state structure with a previous one of ADP-bound Prp43, the authors identify a new element (a hook-loop) that is important for the translocation. Although there is currently no structure for the transition-state of the reaction, the authors include biochemical studies with structure-based mutants that support their proposed mechanism (with the caveat that an important control is missing – see below).

This mechanism proposed for Prp43 is different from that proposed in a previous study reporting the structure of a DEAH/RHA helicase, MLE, bound to RNA and ADP-AlF, e.g. in the transition-state of the reaction. In general, I find interesting that two helicases that are thought to belong to the same family differ in mechanisms. Indeed, not only the translocation mechanism appears to be different, but also the unwinding mechanism and the RNA-binding specificity. Rather than playing down the previous work and boosting their own (with several inappropriate 'first' claims), the authors could elaborate positively on the different mechanisms of translocation, unwinding and specificity of the RHA-like and Prp43-like subfamilies of DEAH helicases.

Specific points:

The authors claim that this is the first structure of a DEAH helicase bound to RNA, and emphasize that the structure provides the first insight into the RNA-binding mechanism of a DEAH-box helicase. However, the RNA binding tunnel of Prp43 is essentially the same as previously reported for MLE. Indeed, the RNA seems to be less well defined in the electron density in the Prp43 structure, with fewer nucleotides and alternative conformations. The claim is therefore inappropriate.

Since the 'hook-loop' has been defined before, the authors should refer to the corresponding Prp43 348-349 residues as 'hook loop 1' and to their newly identified Prp43 180-181 residues as 'hook loop 2', rather than the other way round.

The important control that is currently missing is biochemical evidence that the proposed Prp43 hook-loop binds RNA. If the hook-loop mutant would be impaired in RNA binding, the impairment in unwinding cannot be interpreted as a failure of translocation, but simply as a failure of substrate binding. The authors should measure the RNA-binding affinity of the hook loop mutant and compare it to the wild type. The hook-loop mutant previously implied for translocation in the case of MLE, for example, can bind RNA but cannot unwind.

How is the hook-loop of Prp43 connected with the nucleotide state? In particular, how does it sense the presence or absence of the γ phosphate of ATP?

In Table 3, ctPrp43-IDSB+U16-RNA is missing. The same should be done for the hook loop mutants. In the case of hook loop mutant, RNA activation is clear. But why is the activity going down when RNA is added in the case of the two hook-loop mutants combined? (ctPrpr43-Hook-Loop1&2+ctPfa1-GP).

The authors refer to Prp43 residues 527-640 as the ratchet domain, from the original Hel308 structure paper. From their mechanism, what is the evidence that this domain is ratcheting? If they have no evidence, they should consider a more cautious nomenclature.

The title should more accurately define the content. Since it appears that the translocation mechanisms of Prp43 and MLE are distinct, the title should be more precise: 'Structural insights into the mechanism of the DEAH-box helicase Prp43'.

Reviewer #3:

In this comprehensive analysis of Prp43, the authors present a series of high resolution crystal structures of the protein, in various ligand-bound forms, as it samples conformational states that may play a role in RNA recognition and the function of Prp43 in splicing and translation. The quality of the data is particularly notable, and it is in large part due to the judicious choice of a stable thermophilic variant that can substitute for S.c. Prp43 in splicing assays. In addition, to trapping with specific ligands, the authors captured different conformational states by observing the molecule in different crystal forms. Particularly clever was the use of an engineered disulfide bridge to test the structure and functional attributes of the closed form, as this provided a very nice way to freeze the molecule in single state for solution experiments.

Other positive attributes of the paper are that a dynamic RNA tunnel is clearly observed, and it involves interesting auxiliary domains that are not common to all SF2 proteins. In this way, it does provide insights into the molecular mechanism for Prp43 translocation and binding to RNA. The splicing experiments and other biochemical experiments were of very high quality and these were used effectively to establish structure-function relationships among domains and motifs of the protein.

The major criticism of this work is that the investigators appeared to be functioning "in a bubble", seemingly unaware of the vast crystallographic and biochemical literature on translocation by very closely related proteins. This negatively impacts the paper in several important ways: A) The paper consistently misrepresents the novelty of the findings, seeming to imply that this is the first time translocation has been structurally characterized for a motor protein in the DEXH family, including NS3 from HCV. B) Important similarities and differences from other motors are overlooked, leading to a missed opportunity for understanding the function of individual motor components. Adding to the issue of novelty is that nucleotide-bound Prp43 has been previously structurally characterized in the absence of RNA ligand. So, this is not the first time we have visualized this particular protein. In the opinion of this reviewer, it appears to bind ssRNA in a nonspecific manner not too dissimilar from other SF2 proteins, so it is not entirely clear what we have learned. Without any comparison to related proteins (other than MLE), the authors make it difficult for the reader to ascertain the structural significance of the reported findings.

Specific criticisms:

1) The Introduction, last paragraph, subsection “Crystal structure of Prp43 with bound RNA in an active state”, second paragraph and subsection “Prp43 trapped in the closed conformation is impaired in its helicase activity”, last paragraph, and many other places in the paper say this is the "first insight", "first insight into a genuine DEAH.." These statements, and those throughout the paper, misrepresent the significance of this work, which is certainly not the first structural insight into a member of this family, and it does not provide the first structural or mechanistic insight into translocation by the family members. Full-length HCV NS3 is in the same DEXH family and it has been the subject of two studies in which translocation was captured on polynucleotide substrates. Indeed, in Appleby (JMB, 2011), NS3 opening and closing, together with sequential translocation along polyuridine, was captured in an exhaustive series of crystal structures that provide a complete mechanistic picture of translocation by this family along RNA. Similar work was done by Gu and Rice, but using a DNA oligonucleotide bound to a truncated NS3 variant, which is less directly comparable with the present study. All of that work built upon incisive single molecule experiments that specifically monitored the translocation of NS3 (Myong, Science 2007, and subsequent work from the Bustamante lab) and showed that it follows a mechanistic paradigm originally elucidated by Hopfner for Hel308. Indeed, it has been known for about ten years that SF1 and SF2 helicases translocate by opening and closing the RecA1 and RecA2 domains as a function of ATP binding and release. There are probably 50 papers on this. While HCV NS3 is formally a DECH variant, the related and structurally-characterized Dengue and YFV NS3 proteins are DEAH – so the significance of the third amino acid is not relevant to the argument.

2) Subsection “Prp43 binds RNA in a sequence-independent fashion”, first paragraph. Please compare the RNA interaction network with that shown in Appleby 2011.

3) Subsection “Crystal structure of Prp43 with bound RNA in an active state”, last paragraph and the corresponding figures. While there is much text devoted to the shape of an RNA "tunnel", the authors do not provide figures that really show what this tunnel looks like. Perhaps a space-filling rendering, of the cavity, with the RNA inside in mesh rendering, zoomed in, would help the reader visualize this feature. Again, the authors need to differentiate this cavity from the grooves that are formed upon opening and closing of NS3.

4) Subsection “Crystal structure of Prp43 in an active state”, last paragraph. – The fact that the authors observe several different structures does not mean that the protein is inherently more flexible. They should not infer dynamic information from these results.

5) Subsection “Crystal structure of Prp43 in an active state”, last paragraph. It is difficult for the reader to keep track of which complex is open and which is closed as a function of ATP analog, solvent molecules, or crystallization conditions. Specifically, which ligands trigger the open state? Which stabilize the closed state? The argument is quite opaque as written.

6) Subsection “Prp43 trapped in the closed conformation is impaired in its helicase activity”, first paragraph. What is the significance and meaning of ∆Fs for unwinding? This is not a term that is typically used within the helicase community. It is therefore not clear what it means, how it is quantitated, or what it reveals. Again, the authors seem to be in their own world, not even using the assays or terms that typify the field. Because of this, it is harder to make comparisons with previous work.

7) Subsection “Prp43 trapped in the closed conformation is impaired in its helicase activity”, first paragraph. The fact that opening and closing of the two RecA domains is linked to ATP binding and translocation has been demonstrated for a broad variety of SF1 and SF2 proteins. In addition to Appleby for SF2s, please check out Wei Yang's work on UvrD, or Lohman's collaborative work on Rep or UvrD. Recent work on RIG-I by Marcotrigiano would also be informative.

8) Subsection “The pre- and post-catalytic state of Prp43”, last paragraph. There is no citation provided for previous work on Motif III. There are a number of key papers on a broad variety of SF2 proteins showing that Motif III coordinates ATP and RNA binding. For the most recent one, see Fitzgerald and Pyle, NAR 2016, and citations therein. This shows a structural basis for the involvement of Motif III.

9) Figure 3. Please show primary data on the unwinding time courses. Again, the variables shown on the axes (∆F, for example), are completely unclear, and the data is too limited to evaluate.

10) Figure 5. One cannot tell from this figure the details that are being discussed in the text. Please use better rendering programs to show the networks under discussion. Graphics throughout the paper appear somewhat primitive – which is strange considering the quality of the primary crystallographic data.

---

## [Author Response]

As you will see all were generally positive about the work but also raised some concerns. In particular, it is necessary to test the RNA binding affinity of the hook loop mutants to ensure that the effect on translocation is not simply because of failure of substrate binding. Another point that must be addressed is to put your work in context with work published on DEAH proteins found in flaviviruses. In addition to these points please deal with the other concerns raised as thoroughly as possible.

We agree that it is necessary to show whether the effect of the hook-loop mutation on RNA unwinding is caused by reduced or lost RNA binding. We therefore performed RNA-binding studies by means of fluorescence anisotropy spectroscopy and the results were included into the revised version.

We also added a detailed comparison of our Prp43 structures with the viral NS3 helicase, which is related to the spliceosomal DEAH-box helicases, but also has structurally important differences (e.g. regarding the C-terminal domains and the RNA-binding tunnel).

Reviewer #1: […] While the data and the overall information mark an important contribution to our understanding of RNA biology, especially pre-mRNA splicing, the presentation leaves room for improvement. The writing is verbose, and the manuscript would probably benefit from tightening of the prose. In addition, the figures do not always convey a clear point and are at times hard to follow (examples include Figure 1, Figure 4, Figure 5). It might be helpful to include more schematic representations of the structural features that are highlighted, and a model at the end that emphasizes the conformational changes for the proposed RNA loading process.

As suggested several parts of the manuscript were rephrased to tighten the prose. We improved all figures which now should be much clearer. Schematic representations of the Prp43-RNA interaction interface in Figure 1 and of the ADP/ATP binding in Figure 6 have been added. In Figure 6 the differences between the ATP and ADP states are highlighted. The conformational changes between the ADP/ATP states and as well the conformational changes allowing RNA loading are best visualized by the two supplementary movies, hence no additional schematic figure for the RNA loading was included.

Reviewer #2:

The authors have determined the structure of a DEAH-box helicase, C. thermophilus Prp43, bound to RNA and to ADP-BeF, e.g. in the ground-state of the reaction. Comparing their ground-state structure with a previous one of ADP-bound Prp43, the authors identify a new element (a hook-loop) that is important for the translocation. Although there is currently no structure for the transition-state of the reaction, the authors include biochemical studies with structure-based mutants that support their proposed mechanism (with the caveat that an important control is missing – see below).

This mechanism proposed for Prp43 is different from that proposed in a previous study reporting the structure of a DEAH/RHA helicase, MLE, bound to RNA and ADP-AlF, e.g. in the transition-state of the reaction. In general, I find interesting that two helicases that are thought to belong to the same family differ in mechanisms. Indeed, not only the translocation mechanism appears to be different, but also the unwinding mechanism and the RNA-binding specificity. Rather than playing down the previous work and boosting their own (with several inappropriate 'first' claims), the authors could elaborate positively on the different mechanisms of translocation, unwinding and specificity of the RHA-like and Prp43-like subfamilies of DEAH helicases.

We removed all “first” claims in order to prevent any potential inappropriateness. The differences in mechanisms and substrate specificity between the related RHA-like and DEAH-box helicases were discussed in more detail.

Specific points:

The authors claim that this is the first structure of a DEAH helicase bound to RNA, and emphasize that the structure provides the first insight into the RNA-binding mechanism of a DEAH-box helicase. However, the RNA binding tunnel of Prp43 is essentially the same as previously reported for MLE. Indeed, the RNA seems to be less well defined in the electron density in the Prp43 structure, with fewer nucleotides and alternative conformations. The claim is therefore inappropriate.

According to the classification of helicases, MLE was considered to belong not to the sub-family of (spliceosomal) DEAH helicases but rather to the closely related sub-family of RHA helicases. Due to the very high structural similarity, we followed the suggestion of the reviewer and removed the statement, that our structure represents the first structure of a DEAH-box helicase with bound RNA.

Since the 'hook-loop' has been defined before, the authors should refer to the corresponding Prp43 348-349 residues as 'hook loop 1' and to their newly identified Prp43 180-181 residues as 'hook loop 2', rather than the other way round.

In order to prevent any confusion with regard to the previous definition of the hook loop in the MLE structure, we introduced the name hook-turn for the region 180-181, as it is actually a β-turn, rather than a loop. The term hook loop is now used for the residues 348-350, which corresponds to the hook loop of MLE.

The important control that is currently missing is biochemical evidence that the proposed Prp43 hook-loop binds RNA. If the hook-loop mutant would be impaired in RNA binding, the impairment in unwinding cannot be interpreted as a failure of translocation, but simply as a failure of substrate binding. The authors should measure the RNA-binding affinity of the hook loop mutant and compare it to the wild type. The hook-loop mutant previously implied for translocation in the case of MLE, for example, can bind RNA but cannot unwind.

We agree with the reviewer that the lack of dsRNA unwinding of the hook-turn mutant could be caused by a loss of RNA binding capability. We therefore measured the RNA binding of ctPrp43 and the ctPrp43 hook-turn mutant by means of fluorescence anisotropy. These experiments demonstrated that the binding of RNA to the mutated Prp43 is not impaired, and therefore the loss of unwinding activity is most likely caused by a loss of RNA translocation activity.

These new results were included in the revised version of the manuscript.

How is the hook-loop of Prp43 connected with the nucleotide state? In particular, how does it sense the presence or absence of the γ phosphate of ATP?

The hook-loop (now hook-turn) is not involved in any contacts with the bound ATP molecule or in particular with the γ phosphate. Thus, there is no direct sensing of the nucleotide state by the hook-turn. The movement of the hook-turn is caused by the global rotation of the RecA1 domain. This information is now provided in the revised manuscript.

In Table 3, ctPrp43-IDSB+U16-RNA is missing. The same should be done for the hook loop mutants. In the case of hook loop mutant, RNA activation is clear. But why is the activity going down when RNA is added in the case of the two hook-loop mutants combined? (ctPrpr43-Hook-Loop1&2+ctPfa1-GP).

As suggested by the reviewer, the ATPase activity was also determined for ctPrp43-IDSB+U16-RNA as well as for all other ctPrp43 mutants and these data were added to the revised Table 3.

Regarding the question concerning the reduced RNA-induced stimulation of the double hook-loop mutant the changes in both *K*_M_ and *K*_cat_ have to be addressed. The *K*_M_ value for ATP is lower in the presence of RNA, which means that the affinity of ATP is increased, hence there is still a cooperative effect of RNA and ATP binding. The *K*_cat_ value of the ctPrp43-GP is lower in presence of RNA, which appears to be puzzling. It is very difficult to explain this effect by the currently known structures, due to the fact that the G-patch protein is missing in all structures. As a consequence, the structural basis for the stimulation of ATPase and helicase activities by the G-patch proteins in not understood at all. Hence, one possibility is that the stimulating effect of the G-patch is reduced in case of the double hook-loop mutant upon binding of the RNA. However, this is pure speculation as no corresponding structural or biochemical data are available, and has therefore not been included in the revised version of the manuscript.

The authors refer to Prp43 residues 527-640 as the ratchet domain, from the original Hel308 structure paper. From their mechanism, what is the evidence that this domain is ratcheting? If they have no evidence, they should consider a more cautious nomenclature.

The name ratchet domain was introduced with the first crystal structure analysis of Prp43 by Walbott et al. (2010) and He et al. (2010) and was based on the structural similarity to the DNA helicase Hel308.

We fully agree with the referee that there is no evidence for a ratcheting function of this domain in Prp43, which is consistent with the Prp43-RNA complex structure, as no side chains of this domain bind to the RNA. The structural rearrangements of the C-terminal domains leading to the opening of the RNA-binding tunnel shows that most of the conformational changes occur mainly within the degenerate winged-helix domain, while the conformation of the ratchet domain remains unchanged. Hence, we prefer to keep these as two separate domains instead of using the term HA2 (helicase associated 2) domain, in which both domains are fused. We now denote the “ratchet domain” as “ratchet-like” domain in order to maintain the structural similarity to Hel308.

In the revised version of the manuscript we included a clear statement, that the “ratchet-like” domain has no ratcheting function in Prp43.

The title should more accurately define the content. Since it appears that the translocation mechanisms of Prp43 and MLE are distinct, the title should be more precise: 'Structural insights into the mechanism of the DEAH-box helicase Prp43'.

We changed the title to 'Structural insights into the mechanism of the DEAH-box helicase Prp43' as suggested by the reviewer.

Reviewer #3:

[…] The major criticism of this work is that the investigators appeared to be functioning "in a bubble", seemingly unaware of the vast crystallographic and biochemical literature on translocation by very closely related proteins. This negatively impacts the paper in several important ways: A) The paper consistently misrepresents the novelty of the findings, seeming to imply that this is the first time translocation has been structurally characterized for a motor protein in the DEXH family, including NS3 from HCV. B) Important similarities and differences from other motors are overlooked, leading to a missed opportunity for understanding the function of individual motor components. Adding to the issue of novelty is that nucleotide-bound Prp43 has been previously structurally characterized in the absence of RNA ligand. So, this is not the first time we have visualized this particular protein. In the opinion of this reviewer, it appears to bind ssRNA in a nonspecific manner not too dissimilar from other SF2 proteins, so it is not entirely clear what we have learned. Without any comparison to related proteins (other than MLE), the authors make it difficult for the reader to ascertain the structural significance of the reported findings.

All “first” claims were removed in our revised manuscript to prevent any potential inappropriateness. We added a detailed comparison of Prp43 with HCV NS3 and also deepened the one with MLE in the revised version of the manuscript. Similarities and differences between these helicases were worked out in more detail (for more details see below [specific criticisms point 1]) and we improved the visualization for this purpose (e.g. with the new Figure 1 or the new Figure 1—figure supplement 4). It is true that this is not the first crystal structure of Prp43, but all previous ones contained ADP. There was no structure of Prp43 with bound ATP and/or RNA. The conformation of Prp43 (especially of the C-terminal domains) in the ATP-bound state, which is presented in this study, is clearly different to that of the previously determined Prp43-ADP complexes. The conformational rearrangements of the C-terminal domains of Prp43 in the ATP-bound state have not been unraveled for any other family member of the DEAH/RHA-box family. The available structures of the Prp43-ADP complex (Walbott et al., 2010; He et al., 2010, Tauchert et al., 2016) or of the MLE-U_10_-ADP-AlF_4_- complex (Prabu et al., 2015) all exhibit a conformation of the C-terminal domains which is similar among each other but in striking contrast to that of our new Prp43-ATP-analog complexes.

Specific criticisms:

1) The Introduction, last paragraph, subsection “Crystal structure of Prp43 with bound RNA in an active state”, second paragraph and subsection “Prp43 trapped in the closed conformation is impaired in its helicase activity”, last paragraph, and many other places in the paper say this is the "first insight", "first insight into a genuine DEAH.." These statements, and those throughout the paper, misrepresent the significance of this work, which is certainly not the first structural insight into a member of this family, and it does not provide the first structural or mechanistic insight into translocation by the family members. Full-length HCV NS3 is in the same DEXH family and it has been the subject of two studies in which translocation was captured on polynucleotide substrates. Indeed, in Appleby (JMB, 2011), NS3 opening and closing, together with sequential translocation along polyuridine, was captured in an exhaustive series of crystal structures that provide a complete mechanistic picture of translocation by this family along RNA. Similar work was done by Gu and Rice, but using a DNA oligonucleotide bound to a truncated NS3 variant, which is less directly comparable with the present study. All of that work built upon incisive single molecule experiments that specifically monitored the translocation of NS3 (Myong, Science 2007, and subsequent work from the Bustamante lab) and showed that it follows a mechanistic paradigm originally elucidated by Hopfner for Hel308. Indeed, it has been known for about ten years that SF1 and SF2 helicases translocate by opening and closing the RecA1 and RecA2 domains as a function of ATP binding and release. There are probably 50 papers on this. While HCV NS3 is formally a DECH variant, the related and structurally-characterized Dengue and YFV NS3 proteins are DEAH – so the significance of the third amino acid is not relevant to the argument.

We added a detailed comparison of Prp43 and NS3 structures, which revealed important differences between both families of helicases:

i) We removed all “first” claims in order to prevent any potential inappropriateness.

ii) NS3 is lacking the three C-terminal domains common the DEAH/RHA family, hence the RNA-binding tunnel of Prp43 formed by the RecA domains and the C-terminal domains is not present in NS3.

iii) NS3 contains one different C-terminal domain that is also involved in RNA binding. One important residue (Trp 501) of the NS3 CTD, which is critical for RNA binding and translocation, does not have any equivalent in Prp43 (and all other DEAH/RHA) helicases, indicating significant differences in the mechanism of RNA binding and translocation.

2) Subsection “Prp43 binds RNA in a sequence-independent fashion”, first paragraph. Please compare the RNA interaction network with that shown in Appleby 2011.

We added a figure showing the RNA interaction network (new Figure 1) and we added a comparison with MLE and NS3 as new supplemental figure (Figure 1—figure supplement 4).

*3) Subsection “Crystal structure of Prp43 with bound RNA in an active state”, last paragraph and the corresponding figures. While there is much text devoted to the shape of an RNA "tunnel", the authors do not provide figures that really show what this tunnel looks like. Perhaps a space-filling rendering, of the cavity, with the RNA inside in mesh rendering, zoomed in, would help the reader visualize this feature. Again, the authors need to differentiate this cavity from the grooves that are formed upon opening and closing of NS3.*

We added a new figure (Figure 1) showing the tunnel with the bound RNA.

4) Subsection “Crystal structure of Prp43 in an active state”, last paragraph. The fact that the authors observe several different structures does not mean that the protein is inherently more flexible. They should not infer dynamic information from these results.

Actually, we intended to state that the rearrangement of the C-terminal domains was unexpected, as the several Prp43 structures from different organisms, which were obtained under different crystallization conditions, and as well as the related MLE structure, showed all the same conformation/position of the CTD with respect to the helicase core. It was not our intention to gain dynamic information from crystal structures.

5) Subsection “Crystal structure of Prp43 in an active state”, last paragraph. It is difficult for the reader to keep track of which complex is open and which is closed as a function of ATP analog, solvent molecules, or crystallization conditions. Specifically, which ligands trigger the open state? Which stabilize the closed state? The argument is quite opaque as written.

We used the term ‘open’ or ‘close’ with regard to the position of the C-terminal domains and the accessibility of the RNA-tunnel. The closed state was observed for the RNA/ATP complex and was well for the ADP complex. The open state was observed only for the ATP complex. Both states are independent of the crystallization condition or crystal packing, as for each state at least two independent crystal structures (different space groups, different crystallization conditions) are available.

The crystallographic data suggest that the open state forms upon binding of ATP, while in the presence of RNA the closed state is stabilized. However, crystallization or crystal structures do not provide any information about the equilibrium of the different states in solution.

6) Subsection “Prp43 trapped in the closed conformation is impaired in its helicase activity”, first paragraph. What is the significance and meaning of ∆Fs for unwinding? This is not a term that is typically used within the helicase community. It is therefore not clear what it means, how it is quantitated, or what it reveals. Again, the authors seem to be in their own world, not even using the assays or terms that typify the field. Because of this, it is harder to make comparisons with previous work.

We used the fluorescence-based helicase assay published by Belon and Frick in 2008. In the revised version of manuscript this assay is explained in more detail within the Materials and methods section.

In order to compare the helicase activity of the mutants with respect to wildtype Prp43, we originally just showed the experimental data as ∆F/s. For the revised version these values were converted into an activity (nM/min) corresponding to the amount of dsRNA substrate unwound per minute.

7) Subsection “Prp43 trapped in the closed conformation is impaired in its helicase activity”, first paragraph. The fact that opening and closing of the two RecA domains is linked to ATP binding and translocation has been demonstrated for a broad variety of SF1 and SF2 proteins. In addition to Appleby for SF2s, please check out Wei Yang's work on UvrD, or Lohman's collaborative work on Rep or UvrD. Recent work on RIG-I by Marcotrigiano would also be informative.

We do not refer to the opening of the two RecA-like domains in the course of ATP hydrolysis. In our manuscript, these lines are dedicated to the opening of the C-terminal domains and thus the opening and closing of the RNA binding tunnel.

We completely agree with reviewer that the opening/closing of the RecA domains is a very well-studied feature of SF1 and SF2 proteins. Since this concept is well established, and additionally not the main focus of our manuscript, we don’t want to extend the length of our manuscript by a review and comparison of all helicases.

8) Subsection “The pre- and post-catalytic state of Prp43”, last paragraph. There is no citation provided for previous work on Motif III. There are a number of key papers on a broad variety of SF2 proteins showing that Motif III coordinates ATP and RNA binding. For the most recent one, see Fitzgerald and Pyle, NAR 2016, and citations therein. This shows a structural basis for the involvement of Motif III.

We added to this part of our revised manuscript several citations for the work on Motif III. In our improved version, we now cite Schwer and Meszaros (EMBO J., 2000), Banroques et al. (J. Mol. Biol., 2010) and Fitzgerald et al. (NAR, 2016).

9) Figure 3. Please show primary data on the unwinding time courses. Again, the variables shown on the axes (∆F, for example), are completely unclear, and the data is too limited to evaluate.

We added a supplementary figure (Figure 4—figure supplement 3) showing exemplary primary data of an unwinding time course, which is based on an assay published by Belon and Frick in 2008.

10) Figure 5. One cannot tell from this figure the details that are being discussed in the text. Please use better rendering programs to show the networks under discussion. Graphics throughout the paper appear somewhat primitive – which is strange considering the quality of the primary crystallographic data.

We tried to improve the quality of the figures throughout the whole manuscript. We added a schematic representation of the RNA-interaction network and a depiction of the RNA-binding tunnel (as suggested in your 3^rd^ remark) to Figure 1. Figure 4 (now Figure 5) was extended with a supplementary scheme how this assay system does work. The criticized Figure 5 (Figure 6) was completely redesigned and also a schematic depiction of the interaction network between Prp43 and ATP/ADP was added. The design of Figure 6 (Figure 7) was also slightly improved.